# Fracton magnetohydrodynamics

Marvin Qi,[*] Oliver Hart, Aaron J. Friedman, Rahul Nandkishore, and Andrew Lucas[†]

*Department of Physics and Center for Theory of Quantum Matter,*
*University of Colorado, Boulder CO 80309, USA*

(Dated: August 16, 2022)

We extend recent work on hydrodynamics with *global* multipolar symmetries—known as "fracton hydrodynamics"—to systems in which the multipolar symmetries are *gauged*. We refer to the latter as "fracton magnetohydrodynamics", in analogy to conventional magnetohydrodynamics (MHD), which governs systems with gauged charge conservation. We show that fracton MHD arises naturally from higher-rank Maxwell's equations and in systems with one-form symmetries obeying certain constraints; while we focus on "minimal" higher-rank generalizations of MHD that realize diffusion, our methods may also be used to identify other, more exotic hydrodynamic theories (e.g., with magnetic *sub*diffusion). In contrast to semi-microscopic derivations of MHD, our approach elucidates the origin of the hydrodynamic modes by identifying the corresponding higher-form symmetries. Being rooted in symmetries, the hydrodynamic modes may persist even when the semi-microscopic equations no longer provide an accurate description of the system.

## CONTENTS

---

[*] marvin.qi@colorado.edu
[†] andrew.j.lucas@colorado.edu

## 1.  INTRODUCTION

Recent years have seen an explosion of interest in the dynamics of classical and quantum many-body systems with kinetic constraints [1]. While sufficiently severe constraints and local dynamics [2, 3] may realize strong Hilbert space fragmentation [2–13], preventing the system from relaxing, one generally expects that more mild constraints merely *delay* thermalization due to anomalously slow dynamics [13]. In certain cases [13, 14], the universal properties of these theories can be characterized within the framework of *hydrodynamics*, which is the coarse-grained effective theory of the long-time and long-wavelength dynamics of systems as they relax to equilibrium. As an example, consider interacting charged particles on a lattice, where the Hamiltonian (or quantum circuit, e.g.)  that generates the dynamics conserves both total charge and its dipole moment [15]. The dynamics of thermalization in such a theory is described by a fourth-order, subdiffusive equation [14, 16]; the resulting hydrodynamic universality class characterizing the generic features of this and related constrained models hosts so-called "fracton hydrodynamics," [14, 17–24], as it describes the thermalization of fracton systems [25–35] (systems whose elementary excitations can only move in tandem) as they relax to global equilibrium.

Previous studies of hydrodynamics in fractonic systems explicitly treat the associated multipolar symmetries as *global* [14]. However, to characterize actual fracton phases, one should instead consider *gauged* multipolar symmetries [31–34], which are relevant to proposed realizations of fractons, e.g., in the quantum theory of elasticity [36] and in quantum spin models [37–39]. The latter theories may be regarded as generalized quantum spin liquids that realize an *emergent* compact quantum electrodynamics (QED); there, the underlying local spin model gives rise to emergent electric and magnetic charges, along with gauge fields that obey compact versions of Maxwell's equations [40]. Importantly, the emergent gauge fields in question are typically higher-rank [32], with basic experimental implications that have recently been considered in the literature [41–44]. Note that in any laboratory realization, there will inevitably be dissipative effects that spoil the effective higher-rank electromagnetism, along with nonlinearities in the higher-rank Maxwell's equations. To make clear predictions for experiment, it is thus desirable to consider a formalism that does not treat the microscopic degrees of freedom directly, but instead describes the collective, long-lived degrees of freedom in the system. In generic interacting systems, these long-lived modes are associated with conserved densities (or Goldstone bosons), and their dynamics is dubbed "hydrodynamics"[1] [45–47]. In this work, we develop a hydrodynamic theory of systems with exotic conservation laws and constraints, which give rise to higher-rank variants of electromagnetism.

Somewhat surprisingly, a first-principles derivation of magnetohydrodynamics using one-form symmetries was not done until the past decade [48–53], and so we begin with a review thereof in Sec. 2. The subtlety lies in the fact that the corresponding hydrodynamic theory—magnetohydrodynamics (MHD)—is controlled by an unusual type of symmetry, known as a one-form symmetry. The typical symmetries relevant to conventional hydrodynamics are associated with the constancy in time of the integral over all space of a finite, local density. In $d$ spatial dimensions, an $n$-form symmetry corresponds to the integral of a local density over a manifold with codimension $n$: When

---

[1] Note that our use of the term "hydrodynamics"—the coarse-grained description of systems as they relax to equilibrium—does not require that momentum be conserved, and need not correspond to the Navier-Stokes equations, for example.

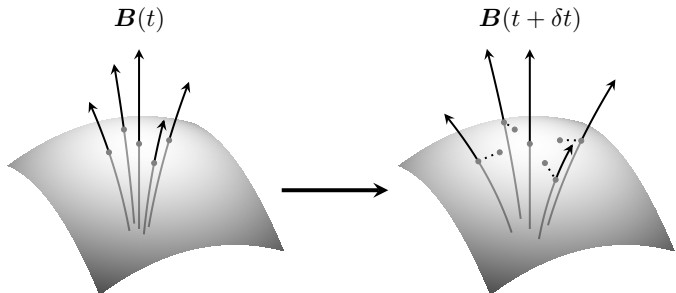

FIG. 1. Schematic illustration of diffusion of magnetic field lines. The field lines' dynamics must preserve the flux of the magnetic field $\boldsymbol{B}$, $\int_S \boldsymbol{B} \cdot \mathrm{d}\boldsymbol{S}$, through any choice of surface, $S$. On the left, there exists a high "concentration" of field lines at the center; on the right, dynamics over time interval $\delta t$ smooths out the local excess of magnetic flux while preserving the total magnetic flux through the surface.

$d = 3$, one-form symmetries correspond to integrals of local densities over two-dimensional surfaces, while two-form symmetries correspond to integrals over one-dimensional curves. The one-form conserved charge in Maxwellian electromagnetism is simply the magnetic flux through arbitrary closed and semi-infinite two-dimensional surfaces. Still, a precise mathematical framework to interpret the hydrodynamics of such conserved charges was only recently developed [54]. In the simplest limit—which describes conventional metals—the only slow (i.e., long-lived) degree of freedom is the magnetic flux density, which diffuses perpendicular to the field direction [55], as depicted in Fig. 1.

This approach, based on higher-form symmetries, has significant conceptual advantages over more familiar semi-microscopic derivations of MHD. Specifically, the symmetry-based approach highlights the underlying symmetries responsible for the observed long-wavelength modes, while also being less limited in its regime of validity than the semi-microscopic approach. For example, in conventional (rank-one) MHD, the semi-microscopic derivation invokes approximate separability of the electromagnetic and matter stress tensors. In the symmetry-based approach [48], one invokes hydrodynamic principles to recover a coarse-grained theory of the long-time and long-wavelength dynamics of the fields in the most interesting and physically relevant regimes, where there may not be a clean separation between the two tensors. This approach also gives predictions for particular limits of conventional $\mathsf{U}(1)$ spin-liquids in which the relevant symmetries are weakly broken. In the case of emergent electromagnetism in *fractonic* spin liquids, the emergent gauge fields are higher rank, leading to additional subtleties and new universality classes. The hydrodynamic description of these higher-rank theories is the subject of this work.

We investigate the simplest example of "fracton magnetohydrodynamics", which arises in rank-two electromagnetism, in Sec. 3. We show that, in the most interesting regime, where electric charges proliferate while magnetic charges do not[2], the higher-rank magnetic field obeys a diffusion equation. This result follows from including electrically charged matter via Ohm's law in the higher-rank generalization of Maxwell's equations in Ref. [32].

---

[2] The systems that we consider generally exhibit electromagnetic duality between electric and magnetic fields and matter. Consequently, analogous results obtain when magnetic charges dominate, instead leading to "electrohydrodynamics", i.e., diffusion of electric field lines.

More interestingly, we interpret this result independently of the semi-microscopic approach. We show that higher-rank MHD naturally arises as a consequence of the theory's one-form symmetry when the conserved density corresponding to that one-form obeys certain global constraints. In Sec. 4 we straightforwardly generalize the construction to higher-rank theories starting either from higher-rank generalizations of Maxwell's equations, or a one-form conserved density combined with certain constraints. We show that at every rank, there exists a self-dual generalization of Maxwell's equations whose universal behavior is described by diffusion of magnetic flux lines. For such theories involving traceless symmetric rank-$n$ tensors, every additional rank introduces two additional diffusing modes, and the diffusion constants at rank $n$ are given by $D\,m^2/n^2$, with $m \in \{1, \ldots, n\}$ and $D = \tau\,c^2$ is the diffusion constant for the rank-one theory, with the relaxation time, $\tau$, a phenomenological parameter. Additionally, all of these theories share the *same* one-form symmetry; we also show how this one-form conserved density and a set of constraints thereupon uniquely determine the rank and form of the generalized Maxwell's equations and the quasinormal mode structure of the corresponding MHD.

In Sec. 5 we discuss a more exotic scenario, in which the densities of electric and magnetic matter in Maxwell's equations themselves carry a vector index. We show how this "vector charge theory" [32], gives rise to *subdiffusive* MHD, and elucidate the combination of one-form symmetries and constraints that lead to subdiffusion, rather than diffusion. Thus, the additional constraints that lead to the fracton magnetohydrodynamics landscape can realize either diffusion or subdiffusion, depending on details of the particular model under consideration, in contrast to systems without multipolar symmetries, which only admit diffusive MHD.

We assume throughout that the systems of interest exist in three spatial dimensions ($\mathbb{R}^3$) and enjoy $\mathsf{SO}(3)$ rotational invariance. The irreducible representations (irreps) of $\mathsf{SO}(3)$ correspond to integer spins $\ell = 0, 1, 2, \ldots$, which we denote according to their dimension: $\mathbf{1}, \mathbf{3}, \mathbf{5}, \ldots$. It is also straightforward to extend our formalism to the reduced point groups relevant to condensed matter realizations, but we relegate such studies to future work.

## 2. ELECTRODYNAMICS TO MAGNETOHYDRODYNAMICS

Before considering systems with multipolar symmetries, we first review the modern hydrodynamic interpretation of standard (rank-one) electromagnetism in terms of "magnetohydrodynamics"— namely, the diffusion of magnetic flux lines in conducting metals (or plasmas) with mobile, electrically charged particles. Starting from Maxwell's equations, we discuss several physical regimes and the corresponding behavior of the electromagnetic fields, and derive magnetic diffusion generically in the presence of electrically charged matter. We interpret these results in the language of hydrodynamics, and argue from more abstract perspectives, following Ref. [48], how a one-form symmetry must arise whenever the charge and current lie in the same irreducible representation of the $\mathsf{SO}(3)$ symmetry. The same approach will be applied to theories that conserve higher multipole moments of charge density in subsequent sections.

## 2.1. Rank-one Maxwell's equations

Standard, rank-one electromagnetism is a field theory describing the behavior of the electric and magnetic fields, $\boldsymbol{E}$ and $\boldsymbol{B}$, in the presence of electrically charged matter with charge density $\rho^{(\mathrm{e})}$ and corresponding current $\boldsymbol{J}^{(\mathrm{e})}$. The dynamics of the $\boldsymbol{E}$ and $\boldsymbol{B}$ fields are governed entirely by Maxwell's equations (2.1). Our discussion applies equally to Maxwell's equations in the vacuum as to *emergent* electromagnetism; while magnetic monopoles do not occur 'naturally', they are to be expected in emergent electrodynamics (and its higher-rank analogues discussed in later sections). Thus, for generality, we allow for a nonzero magnetic charge density, $\rho^{(\mathrm{m})}$, and corresponding current, $\boldsymbol{J}^{(\mathrm{m})}$, in the discussion to follow. In a condensed matter setting, the equations presented in the following section describe, e.g., $\mathsf{U}(1)$ spin liquids with gapless gauge modes and gapped matter [56] (see also Refs. [57–60]), realized perhaps most prominently in quantum spin ice [56, 61, 62].

In standard Cartesian coordinates, Maxwell's equations for the electric and magnetic fields are given by

$$\partial_i E_i = \frac{1}{\varepsilon}\rho^{(\mathrm{e})} \tag{2.1a}$$

$$\partial_i B_i = \mu\,\rho^{(\mathrm{m})} \tag{2.1b}$$

$$\partial_t B_i = -\epsilon_{ijk}\,\partial_j E_k - \mu J_i^{(\mathrm{m})} \tag{2.1c}$$

$$\partial_t E_i = \frac{1}{\mu\,\varepsilon}\epsilon_{ijk}\,\partial_j B_k - \frac{1}{\varepsilon}J_i^{(\mathrm{e})}\ , \tag{2.1d}$$

where $\rho^{(\mathrm{e})}$ and $\rho^{(\mathrm{m})}$ are the electric and magnetic monopole charge densities, respectively, with $J_i^{(\mathrm{e})}$ and $J_i^{(\mathrm{m})}$ the $i$th components of the corresponding currents, and $\mu\,\varepsilon\,c^2 = 1$ defines the speed of light, $c$, in terms of the dielectric permittivity, $\varepsilon$, and the permeability, $\mu$, which characterize the system's linear response to electric and magnetic fields, respectively.

Additionally, because electric (magnetic) charge is locally conserved, charge density and its associated current are related by a continuity equation,

$$\partial_t\rho^{(\mathrm{e})} + \partial_i J_i^{(\mathrm{e})} = 0\,, \tag{2.2}$$

which follows from taking the divergence of Ampère's law (2.1d); the magnetic charge continuity equation takes the same form, and follows from taking the divergence of Faraday's law (2.1c).

Note that magnetic monopoles are not present in standard Maxwellian electrodynamics; thus, in the context of nonemergent electromagnetic systems, such as conventional metals or plasmas in space, one should take $\rho^{(\mathrm{m})} = J_i^{(\mathrm{m})} = 0$. However, in the context of *emergent* electromagnetism, one generally expects to find *both* electric and magnetic charges in generic temperature and parameter regimes. As we will see shortly, for the purposes of realizing interesting hydrodynamics in such materials, it is crucial that a separation of scales exists between the two types of matter so that one of the two species (electric or magnetic) is sufficiently suppressed in density with respect to the complementary species [63].

## 2.2. Hydrodynamic interpretation

An important observation is that Maxwell's equations (2.1) can be regarded as hydrodynamic equations of motion for the electromagnetic fields. In the absence of magnetic matter, Faraday's law (2.1c) can be viewed as a hydrodynamic equation of motion for the $\boldsymbol{B}$ field:

$$\partial_t B_i + \partial_j \left( \epsilon_{ijk} E_k \right) = 0, \tag{2.3}$$

where the second, parenthetical term on the left-hand side plays the role of the hydrodynamic *current* conjugate to the vector-valued conserved density, $\boldsymbol{B}$.

In fact, (2.3) can be recast in the form of a *continuity equation* (e.g., (2.22) in Sec. 2.3) by identifying $B_i$ as a conserved density, and $\epsilon_{ijk} E_k$ as the corresponding current. Then (2.3) takes the standard hydrodynamic form $\partial_t \rho_i + \partial_j J_{ij} = 0$ for a vector charge density, $\rho_i$, corresponding to $B_i$, with the associated current given by $J_{ij} = \epsilon_{ijk} E_k$. Since the conserved density is a [pseudo]vector, the current is rank two: $J_{ij}$ can be interpreted as the current of $B_i$ in the $j$th direction. Effectively, Faraday's law (2.1c) gives an explicit form for the current, obviating the need for a standard constitutive relation in which the currents are expressed in terms of derivative expansions of the conserved densities (in this case, the $\boldsymbol{E}$ and $\boldsymbol{B}$ fields).

A similar procedure can be applied to the $\boldsymbol{E}$ field: Rearranging Ampère's law (2.1d) leads to

$$\partial_t E_i - c^2 \, \epsilon_{ijk} \partial_j B_k = -\frac{1}{\varepsilon} J_i^{(\mathrm{e})}, \tag{2.4}$$

which resembles the magnetic analogue (2.3) but with a [possibly] nonzero source term on the right-hand side. If the source is removed ($\boldsymbol{J}^{(\mathrm{e})} \to 0$), then the electric field is, like the magnetic field, a true conserved density, obeying the standard continuity equation $\partial_t \rho_i + \partial_j J_{ij} = 0$, with $\rho_i \to E_i$ and $J_{ij} \to -c^2 \epsilon_{ijk} B_k$, mirroring (2.3) up to the factor $-c^2$ in defining the conjugate current. When $\boldsymbol{J}^{(\mathrm{e})} \neq 0$, the electric field is no longer conserved[3].

In the presence of *magnetic* charge, Maxwell's equations become fully self dual, and the magnetic field is no longer a conserved density, instead decaying on a time scale set by the conductivity for magnetic monopoles, in accordance with the magnetic analogue of (2.14). In what follows, unless otherwise stated, we will consider Maxwell's equations (2.1) *without* magnetic charge, where the equations are no longer self dual under $\boldsymbol{E} \leftrightarrow \boldsymbol{B}$ (and, correspondingly, e $\leftrightarrow$ m), but the magnetic flux density is exactly conserved.

### 2.2.1. Matter-free limit: The photon

We first consider the matter-free sector, where $\rho^{(\mathrm{e})} = J_i^{(\mathrm{e})} = 0$ (and likewise for magnetic charges). This will serve as a useful point of comparison for the results obtained later on in the context of higher-rank gauge theories. In the absence of charged matter, both the electric and magnetic fields obey a continuity equation of the form $\partial_t \rho_i + \partial_j J_{ij} = 0$ (2.22), where the current corresponding to

---

[3] More precisely, if we apply a Helmholtz decomposition to the electric field, the irrotational (curl-free) component decays on a time scale $\tau^{-1} = \sigma/\varepsilon$, from the argument presented in Eq. (2.14). On the other hand, the solenoidal (divergence-free) component is inextricably tied to the exactly conserved $\boldsymbol{B}$ field (absent magnetic monopoles). From (2.1d), at times $t \gg \tau$, we have a purely solenoidal electric field, $\boldsymbol{E} = \tau c^2 \nabla \times \boldsymbol{B}$, which is locked to the dynamics of $\boldsymbol{B}$, and, hence, diffuses. The "overlap" of $\boldsymbol{E}$ with the diffusing $\boldsymbol{B}$ field vanishes as $k \to 0$, however.

the conserved density $E_i$ is $J_{ij} = -c^2 \epsilon_{ijk} B_k$, and the current conjugate to the conserved density $B_i$ is $J_{ij} = \epsilon_{ijk} E_k$. Taking the curl of Faraday's law (2.1c) and inserting into Ampère's law (2.1d) (and vice versa), e.g., gives the equations of motion

$$\partial_t^2 E_i = c^2 \, \partial^2 E_i \,, \qquad \partial_t^2 B_i = c^2 \, \partial^2 B_i \,, \tag{2.5}$$

where $\partial^2$ is the Laplacian; the expressions above correspond to wave equations for both the electric and magnetic fields, which propagate ballistically at speed $c$. Since Faraday's (2.1c) and Ampère's (2.1d) laws relate $\boldsymbol{E}$ and $\boldsymbol{B}$, the above equations are *not* independent. Taking $E_j \sim E_j\,(\boldsymbol{k}, \omega)\, \mathrm{e}^{\mathrm{i}(k_z z - \omega t)}$ (and similarly for the $\boldsymbol{B}$ field), the system's normal modes are identified as

$$\omega = c|\boldsymbol{k}| \quad \text{for} \quad E_{x,y}\,, \text{ with } \omega \boldsymbol{B} = \boldsymbol{k} \times \boldsymbol{E}\,. \tag{2.6}$$

Because the wave vector, $\boldsymbol{k}$, is taken to be oriented in the $\hat{\boldsymbol{z}}$ direction, (2.6) corresponds to wavelike propagation of the transverse components of the $\boldsymbol{E}$ and $\boldsymbol{B}$ fields along the $\hat{\boldsymbol{z}}$ direction; the two transverse normal modes (i.e., those perpendicular to $\boldsymbol{k} \| \hat{\boldsymbol{z}}$) correspond to the two polarizations of the photon.

A longitudinal photon polarization is forbidden by the matter-free Gauss law constraint (2.1a), whose Fourier transform is given by $k_z E_z = 0$, forcing the longitudinal component of the electric field to vanish. We note that the same holds for the $\boldsymbol{B}$ field, even in the presence of electrically charged matter. The absence of a propagating longitudinal mode can also be justified by identifying a "hidden" conserved quantity, as we discuss in Sec. 2.2.3.

### 2.2.2. The Ohmic regime: Magnetic diffusion

We now consider the hydrodynamic description of the $\boldsymbol{E}$ and $\boldsymbol{B}$ fields in the case most relevant to experiments in electronic materials: the Ohmic regime. There, electrically charged matter obeys Ohm's law, $\boldsymbol{J}^{(\mathrm{e})} = \sigma \boldsymbol{E}$, where $\sigma$ is the Drude conductivity; this limit describes the behavior of mobile charges in conducting materials (including poor conductors), and is the most analytically tractable scenario in which the electric fields decays while the magnetic field remains a good hydrodynamic mode (i.e., a conserved density). This limit can arise in actual electronic materials in the presence of dynamical fields (or in the context of spin liquids, in which case the charges and fields are *emergent*) if there is a large separation of scales between the electric and magnetic conductivities, so that the latter can be ignored [63].

The Ohmic regime is the most generic scenario in which the presence of (electric) charge breaks the conservation of the electric field, $\boldsymbol{E}$, leading to its decay, while the magnetic field remains a good hydrodynamic mode. In Sec. 2.3, we will see that this corresponds to breaking the $\boldsymbol{E}$ field's one-form symmetry while preserving that of the $\boldsymbol{B}$ field.

Microscopically, one expects the matter current, $\boldsymbol{J}^{(\mathrm{e})}$, to be proportional to the force that engenders it—in this case the Lorentz force, $\boldsymbol{F} \propto \boldsymbol{E} + \boldsymbol{v} \times \boldsymbol{B}$. We then introduce the Drude conductivity, $\sigma$ (or equivalently, the relaxation time, $\tau \sim 1/\sigma$), as the coefficient of proportionality $\boldsymbol{J}^{(\mathrm{e})} \sim \sigma \boldsymbol{F}$. Importantly, $\sigma \sim 1/\tau$ is a *phenomenological parameter*, which differs for different materials and must be determined using experiments (likewise, the parameters $\sigma \sim 1/\tau$ introduced

for higher rank theories of electromagnetism will also differ from the $\tau$ discussed here).

Note that we have already implicitly made some restrictions to this phenomenological parameter based on symmetry arguments. Generally speaking, $\sigma$ could be a *matrix*; however, since the system is assumed to exhibit SO(3) rotational invariance, we must construct $\sigma$ out of SO(3)-invariant objects. Since the only compatible such matrix is the identity, $\sigma$ reduces to a scalar. Thus, the current, $\boldsymbol{J}$, is both proportional and parallel to the force that drives it. Additionally, because we are interested in linear response (and linearized hydrodynamics), $\sigma$ must be independent of both the $\boldsymbol{E}$ and $\boldsymbol{B}$ fields. Finally, because the matter velocity field, $\boldsymbol{v}$, has nonzero overlap with other hydrodynamic modes (e.g., the matter current, $\boldsymbol{J}^{(e)}$), the magnetic contribution to the Lorentz force, $\boldsymbol{v} \times \boldsymbol{B}$, is *nonlinear*, and therefore subleading. Hence, we are left with $\boldsymbol{J}^{(e)} = \sigma \boldsymbol{E}$ with $\sigma$ a microscopically determined parameter that is independent of the fields. We do not need to consider $k$ or $\omega$ dependence in $\sigma$, which amount to subleading corrections to hydrodynamics: In generic systems, such corrections are suppressed by the dimensionless combinations of $k\ell_{\mathrm{mfp}}$ or $\omega\tau_{\mathrm{mft}}$ where $\ell_{\mathrm{mfp}}$ ($\tau_{\mathrm{mft}}$) corresponds to a microscopic mean free path (time) for inter-particle scattering. In the context of hydrodynamics, such terms are generically interpreted as higher-derivative corrections to the constitutive relations.

The effect of including a nonvanishing matter current, $\boldsymbol{J}^{(e)} \neq 0$, is to break the conservation of the electric field, $\boldsymbol{E}$, as can be seen upon examination of the right-hand side of Ampère's law (2.4). Following the prescription of quasihydrodynamics [52], we further eschew the Drude conductivity, $\sigma$, in favor of the relaxation time, $\tau$, to recover

$$\partial_t E_i - c^2 \, \epsilon_{ijk} \partial_j B_k = -\frac{1}{\varepsilon} J_i^{(e)} = -\frac{\sigma}{\varepsilon} E_i = -\frac{1}{\tau} E_i \,, \tag{2.7}$$

where, in an Ohmic metal, $\tau \sim \varepsilon/\sigma$, with $\sigma$ the Drude conductivity. Note that the relaxation time for fields, $\tau$, that appears in (2.7) is *not* the same as the scattering time that appears in the Drude conductivity itself.

Having recovered an expression governing the dynamics of the electric field in the Ohmic limit, the hydrodynamic equation of motion for the magnetic flux density is found by taking the curl of (2.7), and inserting the resulting expression into Faraday's law (2.1c), giving

$$\partial_t^2 B_i + \frac{1}{\tau}\partial_t B_i + c^2 \left( \partial_i \partial_j B_j - \partial^2 B_i \right) = 0 \,, \tag{2.8}$$

where we have used the vector calculus identity $\epsilon_{ijk}\epsilon_{kmn}\partial_j\partial_m B_n = \partial_i \partial_j B_j - \partial^2 B_i$ for the double curl, and we note that $\partial_i B_i = 0$ by the magnetic Gauss's law (2.1b). At late times[4], we take $\tau\partial_t \ll 1$, meaning that $\partial_t B_i \gg \tau \partial_t^2 B_i$, resulting in the equation of motion

$$\partial_t B_i = D\, \partial^2 B_i \,, \tag{2.9}$$

corresponding to diffusion of magnetic flux lines, with diffusion constant $D \equiv \tau\, c^2$, depicted schematically in Fig. 1. In Sec. 2.3, we show how this same result (2.9) can be derived from the

---

[4] By late times we mean $t \gg \tau$. At times $t \lesssim \tau$, there exist oscillatory solutions for short wavelength modes satisfying $\tau c k > 1/2$, which decay over a time scale set by $\tau$. At late times, the dominant contribution is from long wavelength modes with $\lambda \gtrsim c\sqrt{t\tau}$, whose dispersion is given by $\omega(\boldsymbol{k}) = -\frac{i}{2\tau} + \frac{i}{2\tau}\sqrt{1 - 4\tau^2 c^2 k^2} = -i\tau c^2 k^2 + O(k^4)$. Analogous arguments are given in Appendix A of Ref. [63].

usual hydrodynamic procedure of constructing the current conjugate to the conserved density, $B_i$, via constitutive relations.

Making use of the generalized divergence theorem, the fact that the $\boldsymbol{B}$ field obeys a standard continuity equation (2.3) implies that the components, $B_i$, are conserved quantities over all space. However, the equations of motion for $B_i$ actually exhibit a much larger set of conservation laws: The total magnetic flux through *any* closed or semi-infinite surface is conserved, as we will see in Sec. 2.3. In fact, this follows already from Faraday's law (2.1c) alone [equivalently, the hydrodynamic continuity equation (2.22)], without the need to appeal to the magnetic Gauss law constraint (2.1b).

From (2.9), the quasinormal modes for the magnetic field corresponding to a wavevector oriented in the $\hat{\boldsymbol{z}}$ direction, $B_i(\boldsymbol{x},t) \propto B_i(\boldsymbol{k},\omega)\,\mathrm{e}^{\mathrm{i}(k_z z - \omega t)}$, are given by

$$\omega = -\mathrm{i}\,\tau\,c^2 k_z^2 \quad \text{for} \quad B_{x,y}\,, \tag{2.10}$$

and, as in (2.6), the longitudinal component, $B_z$, is not a propagating mode since it is constrained to vanish by the [Fourier-transformed] magnetic Gauss's law (2.1b), $k_i B_i = 0$. The transverse components of the field are not constrained by Gauss's law and diffuse, as one would expect from (2.9).

### 2.2.3. *Conservation of fluxes through surfaces*

We now derive the conservation of magnetic flux through arbitrary closed surfaces; this derivation applies equally to electric flux in the absence of electrically charged matter ($\rho^{(\mathrm{e})} = 0$). For simplicity, we restrict our consideration to the magnetic field, where $\rho^{(\mathrm{m})} = 0$ is guaranteed in free space, and assumed in the context of spin liquids.

Note that multiplying the magnetic Gauss's law (2.1b) by an arbitrary, time-independent test function, $\Phi(\boldsymbol{x})$, and integrating over any volume, $D$, still gives zero:

$$\partial_i B_i = 0 \;\rightarrow\; \int_D \mathrm{d}^3 x\, \Phi\, \partial_i B_i = 0\,, \tag{2.11}$$

and using integration by parts, we then find

$$\int_D \mathrm{d}^3 x\, \Phi\, \partial_i B_i = -\int_D \mathrm{d}^3 x\, B_i \left(\partial_i \Phi\right) + \int_{\partial D} \mathrm{d}S\, \Phi\, B_i\, \hat{n}_i\,, \tag{2.12}$$

where $\hat{n}_i$ are the components of the unit vector normal to the surface $\partial D$ (where $\partial D$ is the boundary of the volume, $D$, and $\hat{\boldsymbol{n}}$ points outward from $D$).

Note that applying a total time derivative to this integral also gives zero. Choosing $\Phi = 1$ eliminates the volume integral in (2.12), and applying the total time derivative leads to

$$\frac{\mathrm{d}}{\mathrm{d}t} \int_{\partial D} \mathrm{d}S\, \hat{n}_i\, B_i = 0\,, \tag{2.13}$$

for any domain, $D$, implying that the magnetic flux through the boundary, $\partial D$ of any volume, $D$, is conserved. The same result can be derived alternatively from Faraday's law (2.1c) by considering higher-form symmetries in Sec. 2.3, where we find that the magnetic flux through semi-infinite

surfaces is also conserved.

### 2.2.4. *Absence of diffusion of the conserved electric charge*

Here we explore why Fick's Law of diffusion does not apply to the conserved electric charge, $\rho^{(e)}$. In a conducting medium with conductivity $\sigma$, the charge current is given by Ohm's law, $J_i^{(e)} = \sigma E_i$, whenever there are mobile charges. The continuity equation (2.2) gives rise to exponential relaxation of charge in the bulk of the conducting medium,

$$\partial_t \rho^{(e)} + \sigma \partial_i E_i = \partial_t \rho^{(e)} + \frac{1}{\tau} \rho^{(e)} = 0, \tag{2.14}$$

so that the electric charge density in the bulk decays to zero exponentially on the time scale

$$\frac{1}{\tau} = \frac{\sigma}{\varepsilon}. \tag{2.15}$$

Essentially, the long-range Coulomb interactions easily pulls charges from very far away, and the resulting interaction rapidly screens any test charge placed in the system. The familiar diffusion of conserved charges that one expects for locally interacting charges with a global (rather than gauged) U(1) symmetry is absent here because the charge density is instead driven by the self-generated electric field.

### 2.2.5. *Magnetic charges and regime of validity*

We briefly reinstate magnetic matter in (2.1b) and (2.1c) for the purpose of discussing the regime of validity of the magnetic diffusion recovered in Sec. 2.2.2, which gives rise to a magnetic charge current $\boldsymbol{J}^{(m)} = \sigma_m \boldsymbol{B}$. As discussed in Sec. 2.2.4, the long-ranged nature of the electric (magnetic) fields implies that electric (magnetic) charge density—i.e., the irrotational component of the electric (magnetic) field, $\rho^{(e)} = \varepsilon \partial_i E_i$ ($\rho^{(m)} = \partial_i B_i / \mu$)—decays on a time scale $\tau_e^{-1} = \sigma_e / \varepsilon$ ($\tau_m^{-1} = \sigma_m \mu$). It is, however, the solenoidal components that are responsible for magnetic diffusion. Orienting the wavevector parallel to $\hat{\boldsymbol{z}}$, we find that the $x$ and $y$ components of $\boldsymbol{B}$ in Fourier space satisfy

$$(\mathrm{i}\omega)^2 B_\perp = -c^2 k^2 B_\perp + \mathrm{i}\omega(\tau_m^{-1} + \tau_e^{-1}) B_\perp - \tau_e^{-1} \tau_m^{-1} B_\perp, \tag{2.16}$$

and we assume that there exists a large separation of [time]scales—i.e., $\tau_m \gg \tau_e$. Restricting to wavelengths $ck\tau_e \lesssim 1/2$ (so as to preclude oscillatory solutions), the longest-lived solution to (2.16) is given by

$$\mathrm{i}\,\omega = \frac{1}{2}\left\{\tau_e^{-1} + \tau_m^{-1} - \tau_e^{-1}\tau_m^{-1}\sqrt{(\tau_m - \tau_e)^2 - (2ck\tau_e\tau_m)^2}\right\}. \tag{2.17}$$

In the long-wavelength limit, $ck\tau_e \ll 1$, we find $\mathrm{i}\,\omega = \tau_m^{-1} + \tau_e c^2 k^2 + O(k^4, \tau_e/\tau_m)$; for the diffusion pole to dominate over simple exponential decay (implied by a finite $\tau_m$), there must exist a further restriction on the wavevector, $k$: Specifically, the wavevector regime relevant to magnetic diffusion

is

$$\sqrt{\frac{\tau_e}{\tau_m}} \ll ck\tau_e \ll 1\,. \tag{2.18}$$

If there exists a large separation of scales between the two decay rates, $\tau_e^{-1} \gg \tau_m^{-1}$, then there exists a nonzero window over which magnetic diffusion prevails. Alternatively, in terms of energy scales, the relevant regime is simply

$$\tau_m^{-1} \ll \mathrm{i}\,\omega \ll \tau_e^{-1}\,. \tag{2.19}$$

One scenario that may realize this regime is if the gaps $\Delta_{e(m)}$ to electric and magnetic matter exhibit a [perhaps $O(1)$] separation of scales, as is typically the case in simple models of, e.g., quantum spin ice [62]. Assuming a simple Drude-like expression for the conductivity, the conductivity should scale with the density of the corresponding matter, such that $\tau_{e(m)} \sim e^{\Delta_{e(m)}/T}$ at temperature $T$, leading to $\sqrt{\tau_e/\tau_m} \sim e^{(\Delta_e - \Delta_m)/(2T)}$. At sufficiently low temperatures, $T \lesssim (\Delta_m - \Delta_e)$, we obtain an exponentially large energy window over which magnetic diffusion will be predominant.

## 2.3. One-form symmetries

Having presented a very thorough discussion of the hydrodynamic limit of the conventional Maxwell equations, let us now present a derivation of these properties based on the more modern language of one-form symmetries [48]. We interpret one-form symmetries in hydrodynamic theories as being a consequence of demanding that the conserved density, $\rho_i$, and its corresponding current, $J_{ij}$, *both* realize vector representations of $\mathsf{SO}(3)$, which we denote as the **3**. We then apply these findings to the case of rank-one electromagnetism, and find the results are equivalent to Sec. 2.2.

The standard hydrodynamic equation of motion for a vector-valued density is given by the continuity equation,

$$\partial_t \rho_i + \partial_j J_{ij} = 0\,, \tag{2.20}$$

and, in general, rank-two objects like $J_{ij}$ can be decomposed according to $\mathbf{3} \otimes \mathbf{3} = \mathbf{1} \oplus \mathbf{3} \oplus \mathbf{5}$ [64],

$$J_{ij} = \underbrace{\tfrac{1}{3}\delta_{ij}\,\mathrm{tr}\,[\,J\,]}_{\mathbf{1}} + \underbrace{\epsilon_{ijk}\,J_k}_{\mathbf{3}} + \underbrace{S_{ij}}_{\mathbf{5}}\,, \tag{2.21}$$

where the first term is the trace part, $J_k = \epsilon_{kmn}J_{mn}$ encodes the antisymmetric components of $J_{ij}$, and the tensor $S_{ij} \equiv \left(3J_{ij} + 3J_{ji} - 2\delta_{ij}J_{kk}\right)/6$ encodes the symmetric, traceless part of $J_{ij}$ [64]. In general the current may be in a *reducible* representation of $\mathsf{SO}(3)$, with nonzero overlap with the **1**, **3**, and **5** irreps. Having a current that overlaps with particular irreps (or combinations thereof) gives rise to different hydrodynamic theories with different conservation laws. While one might generally expect the current to have nonzero overlap with all irreps in (2.21), we will focus on the case where $J_{ij}$ is in an *irreducible* representation of $\mathsf{SO}(3)$. This will generally lead to the most conservation laws and the richest structure. A case where the current overlaps with multiple

irreps is discussed in App. A.3.

In particular, in the case where the current, $J_{ij}$ is in the **3** of $\mathsf{SO}(3)$ (the "spin-one" irrep), then the current must be expressible entirely in terms of $\mathsf{SO}(3)$-invariant tensors, and a vector-valued object, $J_k$. In (2.21), that vector object can be extracted from the rank-two object by contracting with the Levi-Civita symbol $\epsilon_{ijk}$. Dropping the **1** and **5** pieces from (2.21), we rewrite (2.20) in terms of $J_k \in \mathbf{3}$ as

$$\partial_t \rho_i + \epsilon_{ijk}\partial_j J_k = 0\,, \tag{2.22}$$

which is precisely the form of Faraday's law (2.1c) (and also Ampère's law (2.1d) in the matter-free limit).

To find the conserved quantities associated with the continuity equation (2.22), consider the putatively conserved quantity

$$\mathcal{Q}[\boldsymbol{f}] \equiv \int_{\mathbb{R}^3} \mathrm{d}^3 x \, f_i\, \rho_i \,, \tag{2.23}$$

where $f_i$ is any vector-valued function of $\boldsymbol{x} \in \mathbb{R}^3$, and $\rho_i$ is a vector-valued density. Since the vector $\boldsymbol{f}$ is time independent, the total time derivative of $\mathcal{Q}[\boldsymbol{f}]$ is given by

$$\frac{\mathrm{d}\mathcal{Q}}{\mathrm{d}t} = \int_{\mathbb{R}^3} \mathrm{d}^3 x \, f_i\partial_t\rho_i = -\int_{\mathbb{R}^3} \mathrm{d}^3 x \, f_i\epsilon_{ijk}\partial_j J_k = \int_{\mathbb{R}^3} \mathrm{d}^3 x \, J_k\epsilon_{ijk}\partial_j f_i - \int_{\partial\mathbb{R}^3} \mathrm{d}S \, f_i\epsilon_{ijk}J_k\hat{n}_j \,, \tag{2.24}$$

where we have invoked the continuity equation (2.22) to write $\partial_t\boldsymbol{\rho}$ in terms of $\boldsymbol{J}$ and then integrated by parts. We require that $\boldsymbol{f}$ and $\boldsymbol{J}$ are well behaved as $|\boldsymbol{x}| \to \infty$, so that the boundary integral above vanishes, giving

$$\frac{\mathrm{d}\mathcal{Q}}{\mathrm{d}t} = \int_{\mathbb{R}^3} \mathrm{d}^3 x \, J_k \left(\epsilon_{ijk}\partial_j f_i\right) \,, \tag{2.25}$$

and we then find that $\mathcal{Q}$ is conserved when the curl of $f_i$ vanishes, i.e.,

$$\epsilon_{ijk}\partial_j f_k = 0\,, \tag{2.26}$$

so that the choice $f_i = \partial_i\varphi$ for some scalar function, $\varphi(\boldsymbol{x})$, leads to a conserved charge, $\mathcal{Q}[\boldsymbol{f}]$, of the form (2.23). One can recover solutions to (2.26) via Helmholtz decomposition of the vector field $\boldsymbol{f}$: Restricting to solutions that are well-behaved as $|\boldsymbol{x}| \to \infty$, the only solution for $f_i$ in Fourier space is one parallel to the wave vector $\boldsymbol{k}$; (2.26) precludes a nonzero "transverse" (or divergence-free, solenoidal) term in the Helmholtz decomposition of $\boldsymbol{f}$, leaving only the parallel (or curl-free, irrotational) component, $f_i = \partial_i\varphi$.

Choosing $\varphi$ to be an indicator function for some finite volume, $V$, i.e.,

$$\varphi_V(\boldsymbol{x}) = \begin{cases} 1 & \boldsymbol{x} \in V \\ 0 & \boldsymbol{x} \notin V \end{cases}\,, \tag{2.27}$$

implies that $f_i = \partial_i\varphi_V(\boldsymbol{x}) = -\delta\left[\boldsymbol{x} \in \partial V\right] \hat{n}_i(\boldsymbol{x})$, where $\delta\left[\ldots\right]$ is a delta function that restricts $\boldsymbol{x}$ to

lie in the boundary, $\partial V$, of the volume, $V$, and $\hat{n}_i$ is the unit vector pointing out of $V$ and normal to $\partial V$. Indicator functions can also be chosen for *semi-infinite* volumes, $V$, such that $\delta [\dots]$ restricts $\boldsymbol{x}$ to some semi-infinite surface (i.e., a boundaryless surface, such as the $xy$ plane, that bounds a semi-infinite region of space).

Essentially, the prescription above gives rise to a conserved charge, $\mathcal{Q}$, that is the integral over a surface, $S$, of the local density. Thus, in addition to conservation of $\rho_i$ over all space, the fact that both $\rho_i$ and its current, $J_{ij} = \epsilon_{ijk} J_k$ are in the **3** of SO(3) leads to a new, one-form conserved charge corresponding to the conservation of the flux of $\rho_i$ through surfaces. That one-form charge is given by

$$\mathcal{Q}_S \equiv \int_S \mathrm{d}S \, \rho_i \hat{n}_i \,, \tag{2.28}$$

where $S$ is an arbitrary closed or semi-infinite surface (we have ignored an overall sign relating solely to the definition of "outside" in the indicator function). The flux through *any* such surface is exactly conserved by the continuity equation (2.22). The importance of $\rho_i$ and its corresponding current, $J_{ij}$, being in the same irrep of SO(3) is that this allows $J_{ij}$ to be expressed in terms of a lower-rank object, $J_k$, and the Levi-Civita tensor $\epsilon_{ijk}$. The appearance of the antisymmetric tensor in $J_{ij} = \epsilon_{ijk} J_k$ guarantees (2.26), and thereby a one-form symmetry. This same argument holds when $\rho_i$ and $J_{ij}$ each carry additional indices, e.g. in the higher-rank theories of electromagnetism considered later.

Returning to the particular case of the electromagnetic fields in the Ohmic regime, we note that Faraday's law (2.1c) is already of the form required to realize a one-form symmetry,

$$\partial_t \rho_i + \epsilon_{ijk} \partial_j J_k \,=\, 0 \,, \tag{2.22}$$

where $\rho_i \to B_i$ is the magnetic field and $J_k = E_k \in \mathbf{3}$ is the electric field. This derives from the ability to write the rank-two current, $J_{ij}$, conjugate to the conserved density, $B_i$, entirely in terms of the $\boldsymbol{E}$ field.

From the hydrodynamic perspective, the current $J_i \in \mathbf{3}$ can be constructed via derivative expansion using the available conserved densities (namely, $\rho_j$) and SO(3)-invariant objects (i.e., $\delta_{jk}$ and $\epsilon_{jk\ell}$). Given that $J_i$ transforms in the vector representation of SO(3), the terms permitted at lowest order are given by

$$J_i \,=\, \alpha \, \rho_i + D \, \epsilon_{ijk} \partial_j \rho_k + \alpha' \partial_i \, \partial_j \rho_j + O(\partial^3) \,, \tag{2.29}$$

where the terms on the right-hand side are the only allowed terms with zero, one, and two derivatives. While the term $J_i \propto \rho_i$ is ostensibly allowed, as both objects belong to the **3**, other considerations preclude $\alpha \neq 0$. For example, if the density $\rho_i$, is odd/even under time reversal, inversion (or parity), or some combination thereof, then the current, $J_i$, must be even/odd under the same transformation; since the term proportional to $\alpha$ in Eq. (2.29) contains no derivatives, thermodynamics forbid any disagreement under either time reversal or inversion. Even allowing for the possibility that time-reversal and/or inversion symmetry are broken microscopically, the effective field theory formalism

of Ref. [65] forbids $\alpha \neq 0$ in general[5]. Hence, we take $\alpha = 0$, so that the leading, symmetry-allowed contribution to the current is $J_i = \epsilon_{ijk}\partial_j\rho_k$, with the latter, $\alpha'$ term in (2.29) subleading (as it contains an extra derivative), and we are left with

$$J_i = D\,\epsilon_{ijk}\partial_j\rho_k\,, \tag{2.30}$$

to leading order, where $D$ is a phenomenological parameter. Using (2.30) for the current in (2.22) gives the equation of motion for the $\boldsymbol{B}$ field ($\rho_i \to B_i$),

$$\partial_t B_i = D\,\partial^2 B_i - D\partial_i\left(\partial_j B_j\right) = D\,\partial^2 B_i\,, \tag{2.31}$$

which is simply the diffusion equation, where Maxwell's equations and Ohm's law allow us to make the identification $D = \tau c^2$. The continuity equation alone (i.e., absent any Gauss law constraint) gives rise to a nondecaying mode. For a density of the form $\rho_i \sim \rho_i(\boldsymbol{k})\mathrm{e}^{\mathrm{i}(k_z z - \omega t)}$, the transverse components diffuse, i.e., $\rho_{x,y}$ decay with rate $\tau c^2 k_z^2$, while the longitudinal component does not decay. In the presence of a Gauss law constraint, the nondecaying longitudinal mode is removed.

## 3. TENSOR ELECTRODYNAMICS AND MAGNETOHYDRODYNAMICS

Here we consider a rank-two theory of electromagnetism analogous to the standard, rank-one theory discussed in Sec. 2. Such theories arise in systems hosting charged matter, which conserve not only electric charge, but also its first moment (i.e., the dipole moment). Regarding the provenience of higher-rank gauge theories in a condensed matter setting, the emergence of higher-rank electromagnetism from microscopic spin-liquid Hamiltonians is discussed at length in Refs. [31, 66–70]. Additionally, certain aspects of these theories are reminiscent of gravity [71], which is also a rank-two theory.

### 3.1. Rank-two Maxwell's equations

The rank-two Maxwell's equations in which the electric and magnetic monopole (i.e., charge) densities are scalars, and the $E$ and $B$ fields are [traceless, symmetric] tensors, take the form [32]

$$\partial_i\partial_j\,E_{ij} = \frac{1}{\varepsilon}\rho^{(\mathrm{e})} \tag{3.1a}$$

$$\partial_i\partial_j\,B_{ij} = \mu\rho^{(\mathrm{m})} \tag{3.1b}$$

$$\partial_t\,B_{ij} = -\frac{1}{2}\left(\epsilon_{ikl}\,\partial_k E_{lj} + \epsilon_{jkl}\,\partial_k E_{li}\right) - \mu J_{ij}^{(\mathrm{m})} \tag{3.1c}$$

$$\partial_t\,E_{ij} = \frac{1}{2\,\mu\,\varepsilon}\left(\epsilon_{ikl}\,\partial_k B_{lj} + \epsilon_{jkl}\,\partial_k B_{li}\right) - \frac{1}{\varepsilon}J_{ij}^{(\mathrm{e})}\,. \tag{3.1d}$$

As in the rank-one case, we recover continuity equations for the electric and magnetic charges by taking the divergence on both indices of Faraday's (3.1c) and Ampère's (3.1d) laws. The continuity

---

[5] If we take $\alpha \neq 0$ as the leading contribution, then the equations of motion become $\partial_t\rho_i = -\alpha\epsilon_{ijk}\partial_j\rho_k$, or $\partial_t^2\rho_i = \alpha^2\partial_i(\partial_j\rho_j) - \alpha^2\partial^2\rho_i$, which is unstable. In the case of MHD, we also have Gauss's law, $\partial_i\rho_i = 0$ (2.1b), and so the equation of motion becomes $\partial_t^2\rho_i = -\alpha^2\partial^2\rho_i$, whose unstable modes are given by $\omega(\boldsymbol{k}) = \pm\mathrm{i}\,\alpha|\boldsymbol{k}|$.

equations are given by

$$\partial_t \rho^{(e/m)} + \partial_i \partial_j J_{ij}^{(e/m)} = 0 \,, \tag{3.2}$$

where the doubled spatial derivative extends the divergence that appears in rank-one theories.

## 3.2. Hydrodynamic interpretation

In analogy to the discussion of rank-one electromagnetism in Sec. 2.2, we recover a hydrodynamic description for the rank-two electric and magnetic fields, $E_{ij}$ and $B_{ij}$, in the absence of their corresponding matter (in the presence of electric matter, the electric field is no longer a conserved density, as in the rank-one case, and likewise for the magnetic field). Because we expect realizations of rank-two quantum electrodynamics (QED) to be *emergent*, we allow for magnetic matter, with magnetic charge density, $\rho^{(m)}$, in (3.1). In the context of, e.g., frustrated magnets, where such higher-rank QED may emerge [31, 33, 34, 66, 67, 69, 70], one generally expects both electric and magnetic quasiparticles, whose densities will both be nonzero at nonzero temperature.

### 3.2.1. Matter-free limit: The photon

In the absence of both electric and magnetic matter, the components, $E_{ij}$ and $B_{ij}$, of both the electric and magnetic field tensors are conserved in accordance with the higher rank continuity equation $\partial_t \rho_{ij} + \partial_k J_{ijk} = 0$. The dispersion relation for the "photon" can once again be derived, e.g., by taking the time derivative of the rank-two Ampère's law (3.1d), then using Faraday's law (3.1c) to express $\partial_t B_{ij}$ in terms of the electric field tensor $E_{ij}$. This results in the wave equation

$$\partial_t^2 E_{ij} = \frac{c^2}{4} \left[ \left( \delta_{in} \partial^2 - \partial_i \partial_n \right) E_{nj} - \epsilon_{ik\ell} \epsilon_{jmn} \partial_k \partial_m E_{n\ell} + i \leftrightarrow j \right] \,, \tag{3.3}$$

where $\mu \, \varepsilon \, c^2 = 1$ defines the [maximum] speed of light, $c$. The equation for $B_{ij}$ assumes the same form, by electromagnetic duality. The system's normal modes can then be found by orienting the the wave vector $\boldsymbol{k}$ along $\hat{\boldsymbol{z}}$. We find four linearly dispersing modes, with two doubly degenerate branches

$$\omega = \frac{c}{2} k_z \times \begin{cases} 1 & E_{xz}, E_{yz} \\ 2 & E_{xy}, E_{xx} = -E_{yy} \end{cases} \,, \tag{3.4}$$

which correspond to ballistic (wavelike) propagation at speed $c/2$ ($E_{xz}$, $E_{yz}$) and speed $c$ ($E_{xy}$, $E_{xx} = -E_{yy}$). In principle, the symmetric, traceless tensor $E_{ij}$ has five independent degrees of freedom. However, one of the resulting five modes is dynamically trivialized by the Gauss law constraints (3.1a) and (3.1b), i.e., the longitudinal components satisfy $k_z^2 E_{zz} = k_z^2 B_{zz} = 0$. Note also that the diagonal elements $E_{xx}$ and $E_{yy}$ appear in the combination $E_{xx} + E_{yy} = 0$, as required by tracelessness.

### 3.2.2. The Ohmic regime: Magnetic diffusion

We now consider the sector in which only one species of charge (electric or magnetic) is present. This may arise, e.g., due to a separation of scales between the gaps for electric versus magnetic matter in materials with emergent QED. The self-dual nature of the traceless scalar charge theory with respect to electric and magnetic fields means that, although we take the limit of vanishing *magnetic* charge density, $\rho^{(m)} = 0$, for concreteness, the results apply equally to the regime of vanishing electric charge density, $\rho^{(e)} = 0$, with the roles of the electric and magnetic fields reversed (up to signs and factors of $c$). The inclusion of both electric and magnetic matter and the corresponding regime of validity is considered in Sec. 3.2.3.

In the absence of magnetic charge, Faraday's law (3.1c) can be interpreted as a continuity equation for the rank-two conserved density $\rho_{ij} \to B_{ij}$. The continuity equation takes the form

$$\partial_t B_{ij} + \frac{1}{2}\left(\epsilon_{ik\ell}\partial_k E_{\ell j} + \epsilon_{jkl}\partial_k E_{\ell i}\right) = 0, \tag{3.5}$$

and, as before, the presence of electric matter in (3.1d) spoils the conservation of the rank-two electric field. Following the prescription of quasihydrodynamics [52], we replace the electric current in (3.1d) according to $J_{ij}^{(e)} = \frac{1}{\tau}E_{ij}$, where $\tau$ is a phenomenological parameter that depends on the material. Since $J_{ij}^{(e)}$ and $E_{ij}$ transform as rank-two tensors, the "electrical conductivity" relating $J_{ij}^{(e)}$ and $E_{ij}$ is constrained by $\mathsf{SO}(3)$ symmetry to be of the form $\sigma_{ijk\ell} = \alpha\delta_{ij}\delta_{k\ell} + \beta\delta_{ik}\delta_{j\ell} + \gamma\delta_{i\ell}\delta_{jk}$. For a traceless symmetric electric field tensor $E_{ij}$, the conductivity is therefore characterized by a single parameter $\varepsilon\tau^{-1} = \beta + \gamma$. Similarly to (2.15) in the rank-one case, we obtain

$$\partial_t E_{ij} - \frac{1}{2}c^2(\epsilon_{ik\ell}\partial_k B_{\ell j} + \epsilon_{jk\ell}\partial_k B_{\ell i}) = -\frac{1}{\tau}E_{ij}, \tag{3.6}$$

and at late times, when $\tau\partial_t \ll 1$, we ignore the time derivative term. Combining this result with Eq. (3.5) gives

$$\partial_t B_{ij} + \frac{1}{4}\tau c^2(3\partial_n\partial_i B_{nj} + 3\partial_n\partial_j B_{ni} - 4\partial^2 B_{ij} - 2\delta_{ij}\partial_m\partial_n B_{mn}) = 0. \tag{3.7}$$

We then seek quasinormal modes corresponding to a wavevector $\boldsymbol{k}$ oriented in the $\hat{\boldsymbol{z}}$ direction, and find four diffusing modes,

$$\omega = -\frac{\mathrm{i}}{4}D\,k_z^2 \times \begin{cases} 1 & B_{xz},\, B_{yz} \\ 4 & B_{xy},\, B_{xx} = -B_{yy} \end{cases}, \tag{3.8}$$

where $D = \tau\,c^2$ is the same diffusion constant identified in the rank-one case (2.9); as with the quasinormal modes for the matter-free sector (3.4), the two branches are distinguished by the propagation speed, $c/2$ versus $c$, and $B_{zz} = 0$.

This mode structure is to be expected based on a general counting argument: A conserved density, $B_{ij} \in \mathbf{5}$ [the traceless, symmetric, rank-two tensor irrep of $\mathsf{SO}(3)$], contains five independent elements, one of which is constrained by Gauss's law (3.1b)—whose Fourier-transform is $k_z^2 B_{zz} = 0$

for propagation in the $\hat{\boldsymbol{z}}$ direction—along with four propagating modes. Thus, $B_{zz}$ is trivially zero by Gauss's law, and tracelessness then requires that $B_{xx} + B_{yy} = 0$. Interestingly, note that fixing the second index of $B_{ij}$ to be $j = z$, gives rise to the *same* three modes recovered in the rank-one case (2.10); additionally, the **5** theory has two additional modes, distinguished by a fourfold suppression of the diffusion constant (each higher rank gives rise to two new propagating modes; the diffusion constants at rank $n$ are given by $D_m = \tau c^2 m^2/n^2$ for $m \in \{1, \ldots, n\}$).

### 3.2.3.  *Magnetic charge and regime of validity*

Including magnetic matter, whose leading effect is to give rise to a current $J_{ij}^{(\mathrm{m})} = \sigma_m B_{ij}$, leads to exponential decay of all rank-two fields at the longest time scales, as was the case for the rank-one theory discussed in Sec. 2.2.5. Specifically, we find that, for well-separated time scales, $\tau_e = \sigma_e/\varepsilon \ll \tau_m = \sigma_m \mu$, the length scales relevant to magnetic diffusion of the higher-rank gauge fields are those satisfying

$$\sqrt{\frac{\tau_e}{\tau_m}} \ll ck\tau_e \ll 1 \,. \tag{3.9}$$

The same condition applies equally to both branches of propagating modes. Above the UV cutoff, there exist remnants of wavelike propagation, and below the IR cutoff, all fields decay exponentially at the same rate, irrespective of the characteristic length scales over which they vary.

## 3.3.   One-form symmetries

The continuity equation (3.5) can be recast in the standard form for a tensor conserved quantity,

$$\partial_t \rho_{ij} + \partial_k J_{ijk} = 0 \,, \tag{3.10}$$

where both the rank-two density, $\rho_{ij}$, and the rank-three current, here $J_{ijk} = \frac{1}{2}(\epsilon_{ik\ell} J_{\ell j} + \epsilon_{jk\ell} J_{\ell i})$, transform in the **5** of $\mathsf{SO}(3)$, corresponding to traceless, symmetric rank-two tensors (i.e., $\rho_{ij}$, $J_{ij} \in \mathbf{5}$). We remind the reader that the vanishing of magnetic charge density can at best only be expected to hold approximately in emergent theories; see Sec. 3.2.3 for a discussion of the length and time scales over which (3.10) provides an accurate description of the dynamics. We find the conserved quantities associated with (3.10) as in the rank-one case (2.23) by considering

$$\mathcal{Q}[f] \equiv \int_{\mathbb{R}^3} \mathrm{d}^3 x \, f_{ij} \rho_{ij} \,, \tag{3.11}$$

where $f_{ij}$ is a traceless, symmetric tensor-valued function of $\boldsymbol{x} \in \mathbb{R}^3$ (note that any components of $f$ not in the **5**—i.e., the trace and the antisymmetric part—cannot contribute to $\mathcal{Q}$, and are therefore not physical). Following the same procedure as used in the rank-one case, we find that $\mathcal{Q}[f]$ is conserved whenever $f_{ij}$ satisfies

$$\epsilon_{\ell ki} \partial_k f_{\ell j} + \epsilon_{\ell kj} \partial_k f_{\ell i} - \frac{2}{3} \delta_{ij} \epsilon_{\ell km} \partial_k f_{\ell m} = 0 \,, \tag{3.12}$$

which derives from the fact that $J_{ijk} = \frac{1}{2}(\epsilon_{ik\ell}J_{\ell j} + \epsilon_{jk\ell}J_{\ell i})$ is in the **5**—i.e., it can be written in terms of the traceless, symmetric tensor $J_{\ell j}$. The last term in (3.12) is identically zero when $f_{ij}$ is symmetric; additionally, the contribution due to a nonzero trace component of $f_{ij} \supset \frac{1}{3}\text{tr}\,[f]\,\delta_{ij}$ from the first two terms will conspire to cancel, since $(\epsilon_{jki} + \epsilon_{ikj})\partial_k\text{tr}\,[f] = 0$. Owing to the antisymmetry of $\epsilon_{ijk}$ in its indices, (3.12) is satisfied precisely when $f_{ij}$ is of the form

$$f_{ij}(\boldsymbol{x}) = \partial_i\partial_j\Phi - \frac{1}{3}\delta_{ij}\partial^2\Phi\,, \tag{3.13}$$

where $\Phi$ is any scalar function of $\boldsymbol{x}$, leading to an infinite family of solutions to (3.12). It is worth noting that (3.13) coincides with the structure of time-independent gauge transformations acting on the vector potential $A_{ij}$, canonically conjugate to $E_{ij}$. This apparent equivalence derives from the self-dual nature of the traceless scalar charge theory—i.e., the derivative and tensor structure of the electric and magnetic Gauss's laws is identical.

The preclusion of other forms of solutions to (3.12) can be justified by appealing to the "scalar-vector-tensor" (SVT) decomposition of $f_{ij}$, which can be viewed intuitively as a Helmholtz decomposition on each index of the rank-two tensor:

$$f_{ij} = f_{ij}^{\parallel} + f_{ij}^{\perp} + f_{ij}^{T}\,, \tag{3.14}$$

where, in Fourier space, the "scalar" component, $f^{\parallel}$, is parallel to the wave vector, $\boldsymbol{k}$, in both indices; the "tensor" component, $f_{ij}^{T}$, is transverse to $\boldsymbol{k}$ in both indices; and the "vector" component, $f_{ij}^{\perp}$, is mixed (being a symmetric sum of two terms that are parallel to $\boldsymbol{k}$ in one index and transverse to $\boldsymbol{k}$ in the other).

The decomposition (3.14) can be realized using the projector,

$$\mathbb{P}_{ij}^{\perp} = \frac{1}{k^2}(k^2\delta_{ij} - k_ik_j) = \frac{1}{k^2}(\epsilon_{ik\ell}k_\ell)(\epsilon_{jkm}k_m)\,, \tag{3.15}$$

which projects onto the subspace orthogonal to $\boldsymbol{k}$, and its complement, $\mathbb{P}^{\parallel} = \mathbb{1} - \mathbb{P}^{\perp}$. Using $\mathbb{P}^{\perp} + \mathbb{P}^{\parallel} = \mathbb{1}$, we resolve the identity on either side of $f$ to recover

$$f = \underbrace{\mathbb{P}^{\parallel}f\mathbb{P}^{\parallel}}_{\text{scalar, }\Phi} + \underbrace{\mathbb{P}^{\parallel}f\mathbb{P}^{\perp} + \mathbb{P}^{\perp}f\mathbb{P}^{\parallel}}_{\text{vector, }v_a} + \underbrace{\mathbb{P}^{\perp}f\mathbb{P}^{\perp}}_{\text{tensor, }T_{ab}} = \left(\begin{array}{c|c}\Phi & v_a \\ \hline v_a & T_{ab}\end{array}\right)\,, \tag{3.16}$$

where, in the last equality, we have written $f$ schematically in a basis in which the wavevector, $\boldsymbol{k}$, locally defines the "parallel" vector $(1,0,0)^T$, so that $\Phi$ lies in the parallel block, $T_{ab}$ lies in the transverse (perpendicular) block, and the components $v_a$ mix between blocks.

The scalar contribution, $k^4\Phi(\boldsymbol{k}) = -k_ik_jf_{ij}$, corresponds to the doubly longitudinal component; the vector, $\boldsymbol{v}(\boldsymbol{k})$, has two independent components[6], and is written in terms of $f_{ij}$ as $k^4v_i(\boldsymbol{k}) = -(\epsilon_{iab}k_a)k_cf_{bc}$; and finally, $k^4T_{ij}(\boldsymbol{k}) = -(\epsilon_{iab}k_a)(\epsilon_{jcd}k_c)f_{bd}$ contains the two remaining degrees of freedom (as $T$ is a symmetric $2 \times 2$ matrix in the subspace orthogonal to $\boldsymbol{k}$, whose trace is fixed[7]).

---

[6] Note that the decomposition (3.14) is not unique, since any components $\boldsymbol{v} \propto \boldsymbol{k}$ will be projected out of $\boldsymbol{v}$.

[7] Much like the vector components[6], $T_{ab}$ is not uniquely determined: The trace is only fixed once the components

Writing out the projectors explicitly—and ensuring that each individual term is traceless and symmetric—gives

$$\underbrace{f_{ij}(\boldsymbol{k})}_{5} = -\underbrace{\left(k_i k_j - \tfrac{1}{3}\delta_{ij}k^2\right)\Phi}_{\text{scalar, 1}} + \underbrace{\left[(\epsilon_{iab}k_a v_b)k_j + i \leftrightarrow j\right]}_{\text{vector, 2}}$$
$$- \underbrace{\left[(\epsilon_{iab}k_a)\left(\epsilon_{jcd}k_c\right)T_{bd} - \delta_{ij}k^2 T_{aa} + \delta_{ij}k_a k_b T_{ab}\right]}_{\text{tensor, 2}}, \tag{3.17}$$

where each term is labelled below according to its role in the SVT decomposition and the number of independent degrees of freedom carried.

Equipped with the decomposition (3.17), we can show that (3.13) is the only solution that leads to conserved quantities of the form (3.11). Inserting (3.17) for $f$ in the relation $(\epsilon_{iab}k_a)f_{bj} = 0$, we see that the scalar term in (3.17) is annihilated independently in each term in (3.12), and therefore a valid solution to $f_{ij}$. However, the "vector" and "tensor" parts of the decomposition (3.17) only satisfy (3.12) if they vanish (this is most apparent from the definitions of $\boldsymbol{v}$ and $T$ in terms of $f_{ij}$). In other words, $(\epsilon_{iab}k_a)f_{bj} = 0$ implies that $f$ must be "parallel" to $\boldsymbol{k}$ in *both* indices since $f_{ij}$ is symmetric, which precludes any contribution from the terms $f^\perp$ and $f^T$ in (3.14) and (3.16), so that only $\Phi$ is nonzero, corresponding to the doubly parallel block in (3.16). Note that the more general equation, $(\epsilon_{iab}k_a)f_{bj} = A_{ij}$ for some antisymmetric tensor $A_{ij}(\boldsymbol{k})$, does not admit new solutions in which the first two terms in (3.12) nontrivially cancel one another (i.e., conspire to cancel without vanishing individually), and we conclude that there are no additional solutions beyond those captured by (3.17)[8].

Having identified (3.13) as the only solutions for $f$ compatible with charges of the form (3.11), in analogy to the rank-one case, we take $\Phi$ to be an indicator function for the volume, $V \subset \mathbb{R}^3$,

$$\Phi_V(\boldsymbol{x}) = \begin{cases} 1 & \boldsymbol{x} \in V \\ 0 & \boldsymbol{x} \notin V \end{cases}, \tag{3.18}$$

we have $f_{ij} = \partial_i \partial_j \Phi_V = -\partial_j \delta\left[\boldsymbol{x} \in \partial V\right]\hat{n}_i(\boldsymbol{x})$. As in the rank-one case, similar indicator functions can be chosen for boundaryless, semi-infinite surfaces, $S$. The conserved quantities associated with choosing $\Phi = \Phi_V$ (3.18) are

$$\mathcal{Q}_S = -\int_{\mathbb{R}^3} \mathrm{d}^3 x \, \rho_{ij}\partial_j \delta[\boldsymbol{x} \in S]\hat{n}_i = \int_S \mathrm{d}S \, \hat{n}_i\left(\partial_j \rho_{ij}\right), \tag{3.19}$$

where $S$ is either the boundary, $\partial V$, of some finite volume, $V$, or a semi-infinite surface, and $\hat{n}$ is the outwardly oriented unit vector normal to $S$. Essentially, the flux of

$$\widetilde{\rho}_i \equiv \partial_j \rho_{ij}, \tag{3.20}$$

through *any* closed or semi-infinite surface is conserved; thus, in systems where the charge and

---

parallel to $\boldsymbol{k}$ have been projected out.

[8] Suppose that $(\epsilon_{iab}k_a)f_{bj} = A_{ij}$, and that the antisymmetric tensor $A_{ij} = \epsilon_{ijk}\lambda_k$ is parametrized in terms of the (for now) arbitrary vector field $\boldsymbol{\lambda}(\boldsymbol{k})$. Since $T_{ij}$ must satisfy $k_i T_{ij} = 0$, and $T_{ij} = (\epsilon_{jcd}k_c)A_{id}$, we find that $k^2\lambda_j - (\boldsymbol{k}\cdot\boldsymbol{\lambda})k_j = 0$, or $\boldsymbol{\lambda} \propto \boldsymbol{k}$. Then $\epsilon_{ijk}k_j f_{k\ell} \propto \epsilon_{i\ell m}k_m$ is solved by $f_{ij} \propto \delta_{ij}$. However, we can also add any function $f_{ij} \propto k_i c_j$, since it belongs to the null space of $\epsilon_{ijk}k_j$. Demanding symmetry of $f_{ij}$ gives the solution $f_{ij} = k_i k_j - \tfrac{1}{3}\delta_{ij}k^2$, which is already captured by setting $\Phi \equiv 1$ in (3.17).

current transform as the **5** of $\mathsf{SO}(3)$, there is an effective one-form symmetry corresponding to the one-form charge, $\widetilde{\rho}_i$ (3.20).

We also note that the rank-two continuity equation (3.5) (or (3.10) in terms of the rank-two magnetic field) expressed in terms of the one-form conserved quantity, $\widetilde{\rho}_i \equiv \partial_j \rho_{ij}$ (3.20) realizes the *rank-one* continuity equation (2.22) for a theory with a one-form symmetry, where both the density and current are in the **3**. Taking the divergence on both sides of (3.5) and using

$$\widetilde{\rho}_i \equiv \partial_j \rho_{ij} \ , \qquad \widetilde{J}_i \equiv \frac{1}{2} \partial_j J_{ij} \, , \tag{3.21}$$

we then find that

$$\partial_t \widetilde{\rho}_i + \epsilon_{ik\ell} \partial_k \widetilde{J}_\ell = 0 \, , \tag{3.22}$$

which has the form of a continuity equation associated with one-form symmetries, and is equivalent to the rank-one Faraday (2.1c) and/or Ampère (2.1d) laws.

Because $\widetilde{\rho}_i$ obeys the continuity equation (3.22), the same arguments invoked in Sec. 2.3 apply—i.e., this describes a theory with a vector-valued conserved density, $\widetilde{\rho}_i$, along with a one-form symmetry associated to the flux of $\widetilde{\rho}_i$ through arbitrary closed or infinite surfaces. Unlike the discussion of Sec. 2.3, however, because $\widetilde{\rho}_i$ arises from taking the divergence of the rank-two conserved density, $\rho_{ij}$, it obeys extra constraints that do not apply to $\rho_i$ in the rank-one case.

The first constraint follows from the fact that $\widetilde{\rho}_i$ is the divergence of a higher-rank object, $\rho_{ij}$, which constrains the total "charge" to be zero,

$$\int \mathrm{d}^3 x \, \widetilde{\rho}_i = \int \mathrm{d}^3 x \, \partial_j \rho_{ij} = 0 \, , \tag{3.23}$$

where the latter equality follows from integration by parts and the fact that $\rho_{ij}$ is well-behaved as $|\boldsymbol{x}| \to \infty$.

The other constraints relate to the "moments" of charge, and derive from properties of $\rho_{ij}$ (specifically, that it's in the **5** of $\mathsf{SO}(3)$). The fact that $\rho_{ij}$ is *traceless* gives the constraint

$$\int \mathrm{d}^3 x \, x_i \, \widetilde{\rho}_i = \int \mathrm{d}^3 x \, x_i \left( \partial_j \rho_{ij} \right) = \int \mathrm{d}^3 x \, \delta_{ij} \, \rho_{ij} = - \int \mathrm{d}^3 x \, \mathrm{tr} \, [\rho] = 0 \, , \tag{3.24}$$

which can be viewed as the "parallel" moment of $\widetilde{\rho}_i$ (again using integration by parts to move the derivative from $\partial_j \rho_{ij}$ to $x_i$). The fact that $\rho_{ij}$ is symmetric (in $i \leftrightarrow j$) gives rise to

$$\int \mathrm{d}^3 x \, \epsilon_{ijk} x_j \, \widetilde{\rho}_k = \int \mathrm{d}^3 x \, \epsilon_{ijk} x_j \left( \partial_\ell \rho_{k\ell} \right) = - \int \mathrm{d}^3 x \, \epsilon_{ijk} \delta_{j\ell} \rho_{k\ell} = - \int \mathrm{d}^3 x \, \epsilon_{ijk} \rho_{jk} = 0 \, , \tag{3.25}$$

which can be viewed as the "transverse" moment of $\widetilde{\rho}_i$, and also relies on integration by parts. Note that, in the context of constraints on $\widetilde{\rho}_i$, "parallel" and "transverse" refer to $\boldsymbol{x}$ in real space; in the context of decomposing $f_{ij}$, these terms refer to $\boldsymbol{k}$ in Fourier space.

The discussion thus far explains how the rank-two Maxwell's equations (3.1) can be viewed as a continuity equation (3.5) for the $B$ field in the Ohmic regime, where both the density, $\rho_{ij} \to B_{ij}$, and current $J_{ij} \to E_{ij}$, are in the **5** of $\mathsf{SO}(3)$ (3.10). This leads to a one-form symmetry corresponding

to conservation of the flux of $\widetilde{\rho}_i = \partial_j \rho_{ij}$ through boundary surfaces. We then note that $\widetilde{\rho}_i$ obeys precisely the continuity equation (3.22) that gives rise to a one-form symmetry corresponding to fluxes of $\rho_i$ in the rank-one case in Sec. 2.3. However, because $\widetilde{\rho}_i$ corresponds to a rank-two conserved quantity in the **5**, it obeys the additional constraints (3.23), (3.24), and (3.25).

We now argue that it is possible to go the other direction: Knowing that a vector-valued conserved density, $\widetilde{\rho}_i$, obeys the one-form symmetric continuity equation (2.22) and respects the above three constraints is sufficient to determine that the underlying theory is second rank, obeys the rank-two continuity equation (3.10), and has the quasinormal modes (3.8) corresponding to rank-two magnetohydrodynamics in the Ohmic regime.

First, the constraint that the total charge vanishes, $\int \mathrm{d}^3 x \, \widetilde{\rho}_i = 0$ (3.23), implies that $\widetilde{\rho}_i$ is the derivative of another object (in this case, that object is higher rank) that need only be well behaved at infinity. Consider a function, $g(x)$ in one dimension, where the Fundamental Theorem of Calculus provides that $g(x) = G'(x)$, with $G$ the antiderivative of $g$. On the circle, e.g., the vanishing of total charge is given by $\int_0^{2\pi} \mathrm{d}\theta \, g(\theta) = G(2\pi) - G(0) = 0$. The zero charge constraint therefore implies that the antiderivative $G(\theta)$ is single-valued. On the real line, we use integration by parts to see

$$\int_{\mathbb{R}} \mathrm{d}x \, g(x) \;=\; G(x)|_{-\infty}^{+\infty} \;\to\; 0\,. \tag{3.26}$$

In this context, the zero-charge constraint implies that the antiderivative, $G(x)$, asymptotically vanishes such that $\lim_{|x|\to\infty} G(x) = 0$ (note that it is unreasonable to require that $G$ be an even function *a priori*, since the constraint is nonlocal). Note that the same considerations also hold for vector-valued $g$. Essentially, the conclusion is that, while *any* smooth function, $g$, can be written in terms of its antiderivative, $G$, the zero-charge constraint further implies that $G(x)$ is well-behaved at infinity (on the real line) or single-valued (on the circle).

The higher-dimensional case is slightly more subtle, as there is no crisp notion of an antiderivative in $\mathbb{R}^d$ for $d > 1$. Nonetheless, we posit that any well-behaved vector-valued function, $g_i \in \mathbb{R}^d$, can be written as the derivative of a higher-rank object, $g_i = \partial_j G_{ij}$ without loss of generality, where the relation between $g$ and $G$ is determined (nonuniquely) by the Helmholtz decomposition[9]. Using higher-dimensional integration by parts, we find

$$\int_{\mathbb{R}^d} \mathrm{d}^3 x \, g_i \;=\; \int_{\mathbb{R}^d} \mathrm{d}^3 x \, \partial_j G_{ij} \;=\; \int_{\partial\mathbb{R}^d} \mathrm{d}S \, G_{ij} \, \hat{n}_j \;\to\; 0\,, \tag{3.27}$$

and thus, the constraint implies $\lim_{|\boldsymbol{x}|\to\infty} |\boldsymbol{x}|^{d-1} \, G_{ij}(\boldsymbol{x}) = 0$, where the factor of $|\boldsymbol{x}|^{d-1}$ is hidden in the measure $\mathrm{d}S$. As in the $d = 1$ case, we see that *generic* vector-valued functions, $g_i \in \mathbb{R}^d$ can be written as $\partial_j G_{ij}$; however, this becomes especially natural when $\int g = 0$, in which case any choice of $G$ that vanishes sufficiently rapidly at infinity and satisfies $\partial_j G_{ij} = g_i$ is valid (on the torus, the constraint is that $G$ is single valued). Effectively, the vanishing of total charge (3.23) immediately implies the existence of a well-behaved, higher-rank object:

$$\int_{\mathbb{R}^3} \mathrm{d}^3 x \, \widetilde{\rho}_i = 0 \quad \Longrightarrow \quad \widetilde{\rho}_i = \partial_j \rho_{ij}\,, \tag{3.28}$$

---

[9] One can view the $G_{ij}$ for a particular $i$ as a vector, where $g_j = \partial_j G_{ij}$ gives the irrotational component, $g = \nabla \cdot G$; the solenoidal component, $\nabla \times G$, is not fixed in this scenario, so the decomposition is not unique.

for some $\rho_{ij}$ that vanishes as $|\boldsymbol{x}| \to \infty$ faster than $1/|\boldsymbol{x}|^2$.

Next, the constraints (3.24) and (3.25) then imply that $\rho_{ij}$ is traceless and symmetric, respectively. We note that, in principle, it is also possible that the trace and antisymmetric components of $\rho_{ij}$ (respectively in the **1** and **3**) are themselves divergences of higher-rank objects—however, as higher-derivative corrections to the definition of $\rho_{ij}$, these subleading terms can safely be ignored[10]. Essentially, at leading order, any nonzero trace component of $\rho_{ij}$ decouples from the hydrodynamic equations governing the components of $\rho_{ij}$ in the **5**; hence these terms are unimportant at the level of hydrodynamics.

Taking the time derivative of (3.25) gives a new constraint on $\widetilde{J}_k$:

$$
\begin{aligned}
0 &= \frac{\mathrm{d}}{\mathrm{d}t} \int \mathrm{d}^3x \, \epsilon_{ijk} \, x_j \, \widetilde{\rho}_i = \int \mathrm{d}^3x \, \epsilon_{ijk} \, x_j \left(-\epsilon_{i\ell m} \partial_\ell \widetilde{J}_m\right) \\
&= \int \mathrm{d}^3x \left(\partial_\ell x_j\right) \left(\epsilon_{ijk}\epsilon_{i\ell m}\right) \widetilde{J}_m = \int \mathrm{d}^3x \left(\epsilon_{ijk}\epsilon_{ijm}\right) \widetilde{J}_m = \int \mathrm{d}^3x \, 2\delta_{km}\widetilde{J}_m \\
&= 2 \int \mathrm{d}^3x \, \widetilde{J}_k = 0,
\end{aligned}
\tag{3.29}
$$

where the second line above relies upon integration by parts; by the same logic used for $\widetilde{\rho}_i$, (3.29) implies that $\widetilde{J}_i = \frac{1}{2}\partial_j J_{ij}$. Substituting the expressions for $\widetilde{\rho}_i$ and $\widetilde{J}_i$ in terms of higher-rank objects into the [ostensibly rank-one] continuity equation (3.22) gives

$$
\partial_t(\partial_j \rho_{ij}) + \frac{1}{2}\epsilon_{ik\ell}\partial_k(\partial_j J_{\ell j}) = 0,
\tag{3.30}
$$

and, extracting an overall $\partial_j$, we determine that the equation of motion for $\rho_{ij}$—by consistency with (3.22) and the rank-one theory—must be of the form

$$
\partial_t \rho_{ij} + \frac{1}{2}\epsilon_{ik\ell}\partial_k J_{\ell j} + \frac{1}{2}\epsilon_{jk\ell}\partial_k \Lambda_{\ell i} = 0,
\tag{3.31}
$$

where the latter term on the left-hand side is the most general term permitted by the constraints on the index structure and is annihilated by $\partial_j$. Symmetry of $\rho_{ij}$ enforces $\Lambda_{\ell i} = J_{\ell i}$ up to subleading corrections (i.e., terms of the form $\partial_\ell \ldots$, that are annihilated by $\epsilon_{jk\ell}\partial_k$), while tracelessness of $\rho_{ij}$ requires that

$$
\partial_k(\epsilon_{ik\ell} J_{\ell i}) = 0,
\tag{3.32}
$$

which implies that either $J_{\ell i}$ is symmetric, or that $\epsilon_{ik\ell}J_{\ell i} = \epsilon_{kmn}\partial_m \Omega_n$. As the latter means that the antisymmetric part of $J_{\ell i}$ appears in the hydrodynamic equation of motion at higher derivative order, we ignore this possibility and take $J_{\ell i}$ to be symmetric. Furthermore, the trace component of $J_{\ell i}$ does not contribute to the hydrodynamic equation of motion, since $\epsilon_{ik\ell}\partial_k(J\delta_{\ell j}) + \epsilon_{jk\ell}\partial_k(J\delta_{\ell i}) = 0$. Hence the continuity equation takes the form of (3.5) with traceless, symmetric charge $\rho_{ij}$ and traceless, symmetric current $J_{ij}$. The normal modes (3.8) follow as a consequence of the hydrodynamic equation of motion.

---

[10] Precisely, any Green's functions of interest for $\rho_{ij}$ would not exhibit any singularities sensitive to the neglected terms—in fact, the neglected terms would be strictly subleading to those included. For example, again orienting $\boldsymbol{k} = k\hat{\boldsymbol{z}}$, $G^{\mathrm{R}}_{\rho_{xy}\rho_{xy}} \sim (Dk^2 - \mathrm{i}\omega)^{-1}k^2 \times (1 + ak^2 + \cdots)$ where the $ak^2$ coefficient comes from total derivative terms we have neglected.

As a quick aside from the present discussion, note that if the tracelessness condition is relaxed such that $\rho_{ij}$ transforms in the $\mathbf{1} \oplus \mathbf{5}$ representation of $\mathsf{SO}(3)$, but (3.23) and (3.25) are still imposed, then the resulting equation of motion is

$$\partial_t \rho_{ij} + \frac{1}{2} \left( \epsilon_{ik\ell} \partial_k J_{\ell j} + \epsilon_{jk\ell} \partial_k J_{\ell i} \right) = 0 \,, \tag{3.33}$$

where, now, $J_{ij}$ now transforms in the *reducible* $\mathbf{3} \oplus \mathbf{5}$ representation of $\mathsf{SO}(3)$—i.e., it is traceless but not symmetric. The resulting rank-two electromagnetic theory then corresponds to a *traceful* electric field with scalar charge density [32], and is not self dual: The electric field tensor, $E_{ij}$, is symmetric but not traceless ($\mathbf{1} \oplus \mathbf{5}$), while the magnetic field tensor, $B_{ij}$, is traceless but not symmetric ($\mathbf{3} \oplus \mathbf{5}$).

Returning to the traceless scalar charge theory (3.1), we recover a constitutive relation for the rank-two current, $J_{ij} \in \mathbf{5}$, via derivative expansion of $\rho_{ij}$ and $\mathsf{SO}(3)$-invariant objects. To low order, this takes the form

$$J_{ij} = \alpha \rho_{ij} + \frac{D}{2} \left( \epsilon_{ik\ell} \partial_k \rho_{\ell j} + \epsilon_{jk\ell} \partial_k \rho_{\ell i} \right) + O(\partial^2) \,, \tag{3.34}$$

and we have neglected subleading contributions at $O(\partial^2)$. The only $\mathsf{SO}(3)$-invariant objects at our disposal are $\delta_{ij}$ and $\epsilon_{ijk}$: Note that the only other zero-derivative term one could write down is $J_{ij} \propto \delta_{ij}\delta_{ab}\rho_{ab} = \delta_{ij}\mathrm{tr}\,[\rho] = 0$; single-derivative terms require use of $\epsilon_{ijk}$, but symmetry of $J_{ij}$ forbids $J_{ij} \propto \epsilon_{ijk}\dots$, and symmetry of $\rho_{ij}$ forbids contracting two indices of the latter with $\epsilon_{ijk}$. In direct analogy to the constitutive relation for the rank-one current (2.29), the term $J_{ij} = \alpha\rho_{ij}$ is forbidden by arguments based on time-reversal symmetry and generic results from effective hydrodynamic theories [65]; most convincingly, $\alpha \neq 0$ leads to unstable (and unphysical) solutions with quasinormal dispersions $\omega = \pm i\alpha|\boldsymbol{k}|, \pm 2i\alpha|\boldsymbol{k}|$. Thus, we take $\alpha = 0$, with the leading contribution proportional to $D$ [identified as $\tau c^2$ in the case of Maxwell's equations (3.1)], giving rise to the quasinormal modes (3.8).

## 4. STANDARD HIGHER-RANK GENERALIZATIONS OF ELECTROMAGNETISM

Having explained the "higher-form symmetry" formulation of magnetohydrodynamics for both conventional (rank-one) electromagnetism and its rank-two (fractonic) generalization, we now turn to generalizing to traceless symmetric rank-$n$ theories. In the interest of simplicity, we make the "standard" assumption that the electric and magnetic charge densities are scalars, that the generalized Maxwell equations are self dual, and that the $E$ and $B$ fields are both in the same irrep of $\mathsf{SO}(3)$. While other choices exist, and may lead to exotic hydrodynamic theories (see, e.g., Sec. 5), the theories that obtain from the aforementioned restrictions are physically most similar to rank-one MHD, and thus we refer to this class of higher-rank theories as "standard". Microscopic Hamiltonians that realize such rank-$n$ theories can be constructed using ideas analogous to those presented in Refs. [32, 41, 68]. For completeness, we also discuss a concrete lattice model realization in the next section.

To generalize the results of the preceding sections to rank-$n$ theories of electromagnetism, it will first prove convenient to define appropriate generalizations of the divergence and curl operators

that appear in the higher-rank extensions of Maxwell's equations. The action of these operators on some totally symmetric tensor $A_{i_1,\ldots,i_n}$ is given explicitly by

$$\nabla^{(n)} \cdot A \equiv \partial_{i_1} \partial_{i_2} \ldots \partial_{i_n} A_{i_1,i_2,\ldots,i_n} \tag{4.1a}$$

$$(\widetilde{\nabla} \times A)_{i_1,\ldots,i_n} \equiv \frac{1}{n} \left( \epsilon_{i_1 k \ell} \partial_k A_{\ell,i_2,\ldots,i_n} + \ldots \right), \tag{4.1b}$$

where the generalized $n$-fold divergence, $\nabla^{(n)} \cdot A$, amounts to taking the divergence of each index of $A$ individually, and the symmetrized multi-index curl, $\widetilde{\nabla} \times A$, is given simply by applying the usual curl, $\epsilon_{i_m k \ell} \partial_k$, to each index, $i_m$, of $A$, and taking the average, so that the resulting object remains symmetric in all indices. The former has $n$ derivatives, while the latter contains only one derivative for all $n$; these generalized derivative operators reduce to the standard divergence and curl for $n = 1$. The standard vector calculus identity, $\text{div}(\text{curl}\,A) = 0$, also applies to the rank-$n$ variants (4.1). In the discussion to follow, we also make use of the multi-index $I_n \equiv \{i_1, i_2, \ldots, i_n\}$, to unencumber notation when working with higher-rank indices. Using this language, the generalized divergence in (4.1a), e.g., can alternatively be written $\partial_{I_n} A_{I_n} \equiv \nabla^{(n)} \cdot A$.

## 4.1. Rank-$n$ Maxwell's equations

Making use of the generalized derivatives in (4.1), the natural generalization of the rank-one (2.1) and rank-two (3.1) Maxwell's equations to rank-$n$ fields and currents with scalar electric and magnetic charge is

$$\nabla^{(n)} \cdot E = \frac{1}{\varepsilon} \rho^{(e)} \tag{4.2a}$$

$$\nabla^{(n)} \cdot B = \mu \rho^{(m)} \tag{4.2b}$$

$$\partial_t B = -\widetilde{\nabla} \times E - \mu J^{(m)} \tag{4.2c}$$

$$\partial_t E = \frac{1}{\mu \varepsilon} \widetilde{\nabla} \times B - \frac{1}{\varepsilon} J^{(e)}, \tag{4.2d}$$

where both the rank-$n$ fields $E_{I_n}$, $B_{I_n}$, and the current $J_{I_n}$, are fully symmetric and traceless. As in previous sections, we take $c$ to be defined by $c^2 = (\mu\varepsilon)^{-1}$ despite the presence of multiple photon branches in the matter free case, each with its own "speed of light". Taking the generalized divergence over all $n$ indices of either Faraday's (4.2c) or Ampère's (4.2d) law gives rise to the appropriate continuity equation for the scalar charge density $\rho^{(e)}$ or $\rho^{(m)}$, respectively

$$\partial_t \rho^{(e/m)} + \nabla^{(n)} \cdot J^{(e/m)} = 0. \tag{4.3}$$

## 4.2. Microscopic Hamiltonians

For completeness, we provide a sketch of how microscopic lattice models that realize higher-rank gauge theories can be constructed systematically. The construction follows closely the approach taken in, e.g., Refs. [32, 41, 68]; the Hilbert space is constructed from rotor degrees of freedom that live on the sites of a face-centered cubic lattice with an additional lattice site at the center (as

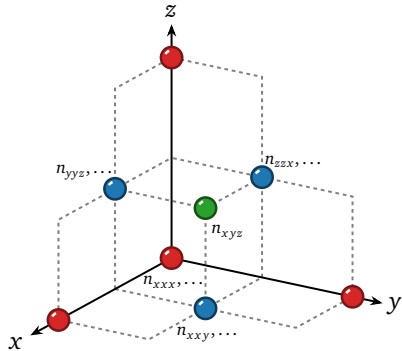

FIG. 2. Schematic depiction of the sites that comprise the microscopic lattice Hamiltonian that realizes a rank-three theory. The red sites, which live on the sites of a simple cubic lattice, indexed by $\boldsymbol{r}$, host the three rotors $n_{xxx}$, $n_{yyy}$, and $n_{zzz}$. The blue sites, which live at the centers of the plaquettes of the cubic lattice, host two rotors each. For example, $n_{xxy}$ and $n_{yxx}$ in the $xy$ plane. Finally, the green site, which lives on the sites of the dual cubic lattice, hosts a single rotor: $n_{xyz}$.

shown in Fig. 2). The Hilbert space of each individual rotor degree of freedom is spanned by angular momentum eigenstates, with integer angular momentum quantum number $n$, and approximates that of a large-$S$ quantum spin under the mapping $\hat{S}^z \sim n - 1/2$ and $\hat{S}^\pm \sim e^{\pm i\theta}$. While we focus on realizing the symmetric scalar charge theory (whose traceless variant will be the focus of the following section), other theories can be constructed using the same Hilbert space in the presence of different Hamiltonians that give rise to different Gauss law constraints.

As is depicted in Fig. 2, the unit cell consists of 10 rotors: Three rotors, $n_{xxx}(\boldsymbol{r})$, $n_{yyy}(\boldsymbol{r})$, and $n_{zzz}(\boldsymbol{r})$ are placed at the vertices of a simple cubic lattice, whose sites are located at positions $\boldsymbol{r}$; two rotors are then placed at the center of each plaquette, e.g., $n_{xxy}(\boldsymbol{r})$ and $n_{xyy}(\boldsymbol{r})$ are situated in the $xy$ plane; finally, a single rotor, $n_{xyz}(\boldsymbol{r})$, is placed on the corresponding vertex of the dual simple cubic lattice. Note that rotors are labeled by the unit cell to which they belong, rather than their actual locations within the unit cell.

The charge-free electric Gauss' law, $\partial_i\partial_j\partial_k E_{ijk} = 0$, can be reproduced in the rotor language by replacing the derivatives that appear in the continuum theory with the corresponding finite difference operators. One possible discretization uses the one-sided derivative:

$$\partial_i\partial_j\partial_k E_{ijk} \leftrightarrow \sum_{ijk} \Big[ n_{ijk}(\boldsymbol{r}) + n_{ijk}(\boldsymbol{r} + \boldsymbol{e}_i) + n_{ijk}(\boldsymbol{r} + \boldsymbol{e}_j) + n_{ijk}(\boldsymbol{r} + \boldsymbol{e}_k) +$$
$$n_{ijk}(\boldsymbol{r} + \boldsymbol{e}_i + \boldsymbol{e}_j) + n_{ijk}(\boldsymbol{r} + \boldsymbol{e}_j + \boldsymbol{e}_k) + n_{ijk}(\boldsymbol{r} + \boldsymbol{e}_k + \boldsymbol{e}_i) +$$
$$n_{ijk}(\boldsymbol{r} + \boldsymbol{e}_i + \boldsymbol{e}_j + \boldsymbol{e}_k) \Big].$$

The lattice variant of the electric field tensor is obtained from the rotor variables $n_{ijk}$ via an alternating sign $E_{ijk}(\boldsymbol{r}) = (-1)^{r_x + r_y + r_z} n_{ijk}(\boldsymbol{r})$ (and, similarly, the vector potential is obtained from the operator $\theta$, conjugate to $n$, via $A_{ijk} \sim (-1)^{r_x + r_y + r_z} \theta_{ijk}$). The corresponding contribution to the Hamiltonian then takes the form of a soft [quadratic] constraint $\sim \lambda(\partial_i\partial_j\partial_k E_{ijk})^2$, which ensures that the ground state of the model is free of charge, $\partial_i\partial_j\partial_k E_{ijk} = 0$, in the limit $\lambda \to \infty$, when the constraint is exact. The theory can then be endowed with dynamics by writing down additional terms that commute with, and hence preserve, the Gauss law constraint (but mix states within a fixed total charge sector). For further details on how such "$E^2 + B^2$" terms can be constructed,

we refer the reader to Refs. [32, 66, 67]. Different theories are obtained by writing down different Gauss law constraints, which select the configurations of $E_{ijk}$ that are energetically penalized.

## 4.3. Hydrodynamic interpretation

As in previous sections, both the electric and magnetic fields are conserved in the absence of their corresponding matter. In the following we derive the general mode structure for the photon in the absence of both species of matter, and then derive diffusion of the rank-$n$ magnetic field when only electric matter is present. Since the higher rank theories that we discuss are expected to be emergent, both species of matter are generally expected to be present with nonzero density at nonzero temperatures. The energy and length scales over which magnetic diffusion prevails, laid out in Sec. 2.2.5 for the rank-one case, also describe the regime of validity of the rank-$n$ theories.

### 4.3.1. Matter free limit: The photon

To discuss the normal modes of the rank-$n$ theory in the absence of both species of matter, we will find it convenient to introduce a new notation for the components of symmetric tensors such as the electric field $E_{I_n}$. Since the tensor is fully symmetric by assumption, each component of the field can be indexed by the number of $x$'s, $y$'s and $z$'s that it contains, i.e., $E_{n_x,n_y,n_z} \equiv E_{\{n_i\}}$ with $n_i \geq 0$ and $\sum_{i=1}^{d} n_i = n$. For instance, $E_{xxy} \to E_{2,1,0}$ in the simplified notation. Taking the time derivative of Ampère's law (4.2d) and making use of Faraday's law to replace $\partial_t B$, we find the following equation of motion for the rank-$n$ $E$ field, having oriented the wave vector parallel to $\hat{z}$

$$\omega^2 E_{\{n_i\}} = \frac{c^2 k_z^2}{n^2} \left[ (n_x + n_y + 2n_x n_y) E_{\{n_i\}} - n_y(n_y - 1) E_{n_x+2,n_y-2,n_z} - n_x(n_x - 1) E_{n_x-2,n_y+2,n_z} \right].$$
(4.4)

It can be verified explicitly that the equation of motion preserves the tracelessness constraint $E_{m_x+2,m_y,m_z} + E_{m_x,m_y+2,m_z} + E_{m_x,m_y,m_z+2} = 0$ (with $m_x + m_y + m_z + 2 = n$), as it must. While it may appear from (4.4) that sectors with fixed $n_z$ are not coupled by the dynamics, this is not the case once the tracelessness constraints are taken into account. For the case $n = 3$, the theory has six ballistically propagating modes. The rank-three tensor has $d^n = 27$ components, of which only $\binom{d+n-1}{n} = 10$ are independent by symmetry, and a further $\binom{d+n-3}{n-2} = 3$ are removed by the tracelessness constraint. This leaves us with the following seven modes

$$\omega = \frac{c}{3} k_z \times \begin{cases} 1 & E_{yzz} = -4E_{xxy}, \ E_{xzz} = -4E_{xyy} \\ 2 & E_{xyz}, \ E_{xxz} = -E_{yyz} \\ 3 & E_{xyy}, \ E_{xxy} \\ 0 & E_{zzz} \end{cases},$$
(4.5)

where the values of the field components $E_{xxx}$, $E_{yyy}$ and $E_{zzz}$ are determined by the tracelessness constraints. The longitudinal mode $E_{zzz}$ is then removed by Gauss' law, leaving the six modes that are not 3-fold parallel to $\hat{z}$. More generally, rank-$n$ traceless symmetric tensors in $d = 3$ possess $2n + 1$ independent components, giving rise to $2n$ dynamical modes and one nondecaying

mode $E_{0,0,n}$. Eliminating this nondynamical mode using Gauss's law, there are $2n$ dynamical modes grouped into $n$ two-fold degenerate branches with speeds in the range $[c/n, c]$ with dispersion relations $\omega = ck_z m/n$ for $m = 1, 2, \ldots, n$.

### 4.3.2. The Ohmic regime: Magnetic diffusion

We now permit nonzero electric charge density and current while maintaining a vanishing density of magnetic charges, $\rho^{(\mathrm{m})}$. In this limit, Faraday's law (4.2c) may be interpreted as a continuity equation for the rank-$n$ locally conserved density $\rho_{I_n} = B_{I_n}$. Specifically,

$$\partial_t B + \widetilde{\nabla} \times E = 0\,, \tag{4.6}$$

may be written $\partial_t \rho_{I_n} + \partial_{i_{n+1}} J_{I_n, i_{n+1}} = 0$ with a rank $n+1$ current. On the other hand, the continuity equation for the electric field is sourced by a nonvanishing current $J^{(\mathrm{e})}$. Following the prescription of quasihydrodynamics, the exact conservation of $E$ is broken by introducing a time scale $\tau$

$$\partial_t E + \widetilde{\nabla} \times B = -\frac{1}{\tau} E\,. \tag{4.7}$$

The time scale, $\tau$, characterizes the decay of the $n$-fold longitudinal component of the electric field (i.e., the electric charge density). As explained in detail in Sec. 2.2.2, this procedure can alternatively be thought of as imposing an Ohm's law relationship between the current and the field that drives it; in this case $J^{(\mathrm{e})}_{I_n} = \sigma_e E_{I_n}$ with a scalar conductivity $\sigma_e$, which describes the system's linear response at sufficiently long length and time scales. By analogy with the discussion above Eq. (3.6), this is the most general form of the conductivity permitted by $\mathsf{SO}(3)$ symmetry: any rank-$2n$ $\mathsf{SO}(3)$-invariant tensor is expressible in terms of products of $\delta_{ij}$. All contributions from $\delta_{ij}$ with both $i$ and $j$ contracted with $E_{I_n}$ vanish by virtue of tracelessness of $E_{I_n}$. A nonvanishing contribution therefore has $i$ associated with $J$ and $j$ contracted with $E$ (or vice versa), for all terms in the product, giving rise to a contribution $J^{(\mathrm{e})} \supset E_{\pi(I_n)}$, for some permutation $\pi$ of the indices $\{i_1, \ldots, i_n\}$; since $E_{I_n}$ is totally symmetric, we obtain $J^{(\mathrm{e})}_{I_n} \propto E_{I_n}$.

At sufficiently long times, when $\tau \partial_t \ll 1$, we drop the time derivative in (4.7) and substitute the resulting relationship between $E$ and $B$ fields into Faraday's law (4.2c). This leads to an equation of motion analogous to (4.4). Defining $D = \tau c^2$ in accordance with (2.15)

$$\mathrm{i}\omega B_{\{n_i\}} = \frac{Dk_z^2}{n^2}\left[(n_x + n_y + 2n_x n_y)B_{\{n_i\}} - n_y(n_y - 1)B_{n_x+2,n_y-2,n_z} - n_x(n_x - 1)B_{n_x-2,n_y+2,n_z}\right], \tag{4.8}$$

where the $B$ field satisfies the tracelessness constraints $B_{m_x+2,m_y,m_z} + B_{m_x,m_y+2,m_z} + B_{m_x,m_y,m_z+2} = 0$ (with $m_x + m_y + m_z + 2 = n$). The mode structure mirrors that of the matter free case. For instance, for the $n = 3$ theory there are six dynamical modes, while the 3-fold longitudinal mode is

unable to decay

$$\omega = -\frac{\mathrm{i}}{9}Dk_z^2 \times \begin{cases} 1 & B_{yzz} = -4B_{xxy}, \ B_{xzz} = -4B_{xyy} \\ 4 & B_{xyz}, \ B_{xxz} = -B_{yyz} \\ 9 & B_{xyy}, \ B_{xxy} \\ 0 & B_{zzz} \end{cases}, \tag{4.9}$$

where the values of the field components $B_{xxx}$, $B_{yyy}$ and $B_{zzz}$ are determined by the tracelessness constraints. The longitudinal mode $B_{zzz}$ is then removed by Gauss' law, leaving the six transverse modes, with diffusion constants $D_m = Dm^2/n^2$ for $m = 1, \ldots, n$, with each branch doubly degenerate. This normal mode structure also generalizes to $n > 3$.

In the presence of magnetic charge, the regime of validity of (4.9), i.e., the length and time scales over which magnetic diffusion occurs, is identical to the rank-one and rank-two theories, presented in Secs. 2.2.5 and 3.2.3, respectively.

## 4.4. One-form symmetries

In the absence of magnetic currents, Faraday's law (4.2c) can be recast as a continuity equation for the conserved density $\rho_{I_n} = B_{I_n}$

$$\partial_t \rho + \widetilde{\nabla} \times J = 0. \tag{4.10}$$

Following the Secs. 2.3 and 3.3, we consider the putatively conserved quantity

$$\mathcal{Q}[f] \equiv \int_{\mathbb{R}^3} \mathrm{d}^3x \, f_{I_n} \rho_{I_n} \,, \tag{4.11}$$

where $f_{I_n}$ can be chosen to be traceless and symmetric. In order for $\mathcal{Q}[f]$ to be conserved, i.e., $\dot{\mathcal{Q}}[f] = 0$, the tensor-valued $f_{I_n}$ satisfies

$$\epsilon_{mk(i_1|}\partial_k f_{m|i_2\ldots i_n)} = 0\,, \tag{4.12}$$

which follows from integrating (4.11) by parts and utilizing the symmetry properties of the $J_{I_n}$, which is also traceless and symmetric. The parentheses indicate symmetrization over the surrounded indices, i.e., $T_{(i_1,\ldots,i_n)} = \frac{1}{n!}\sum_{\pi \in S_n} T_{\pi(i_1,\ldots,i_n)}$, where $\pi$ is a permutation belonging to the symmetric group $S_n$. The vertical bars denote indices that are to be excluded from the symmetrization procedure. We have omitted terms that vanish due to the assumed symmetry of $f_{i_1,\ldots,i_n} = f_{(i_1,\ldots,i_n)}$. Accounting for the tracelessness of $f$, the appropriate solution to the above is

$$f_{I_n} = \partial_{i_1}\cdots\partial_{i_n}\Phi - \frac{\binom{n}{2}}{2n-1}\delta_{(i_1,i_2}\partial_{i_3}\cdots\partial_{i_n)}\partial^2\Phi + \frac{3\binom{n}{4}}{(2n-1)(2n-3)}\delta_{(i_1,i_2}\delta_{i_3,i_4}\partial_{i_5}\cdots\partial_{i_n)}\partial^4\Phi + \ldots \tag{4.13}$$

where the second and third terms[11] on the right-hand side progressively remove the trace part of $f$. Taking the curl on any of $f_{I_n}$'s indices vanishes by antisymmetry of $\epsilon_{ijk}$. Choosing the same indicator functions as, e.g., (3.18), we identify the conserved quantities as

$$\mathcal{Q}_S = -\int_{\mathbb{R}^3} \mathrm{d}^3 x \, \rho_{i_1,\dots,i_n} \partial_{i_2} \cdots \partial_{i_n} \delta[\boldsymbol{x} \in S] \hat{n}_{i_1} = (-1)^n \int_S \mathrm{d}S \, \partial_{i_2} \cdots \partial_{i_n} \rho_{i_1,\dots,i_n} \hat{n}_{i_1} \ , \tag{4.14}$$

where $S = \partial V$ for some volume $V$. That is, the flux of the object $\widetilde{\rho}_{i_1} \equiv \partial_{i_2} \cdots \partial_{i_n} \rho_{i_1,\dots,i_n}$ through any closed or semi-infinite surface is conserved by the rank-$n$ continuity equation (4.10). Hence, in systems whose charge and current are both symmetric traceless tensors of the same rank [in the $(2n+1)$-dimensional irrep of $\mathsf{SO}(3)$], there is an effective one-form symmetry whose charge is given by $\widetilde{\rho}_{i_1} \equiv \partial_{i_2} \cdots \partial_{i_n} \rho_{i_1,\dots,i_n}$. This explains the presence of the nondecaying mode in (4.5), since for a surface $\Sigma$, the rank-$n$ theory conserves

$$\mathcal{Q}_\Sigma = \rho_{i_1,i_2,\dots,i_n}(t)(k_{i_2} \cdots k_{i_n}) \int_\Sigma \mathrm{d}S \, n_{i_1} e^{i\boldsymbol{k}\cdot\boldsymbol{x}} = \rho_{iz\cdots z}(t) k_z^{n-1} \int_\Sigma \mathrm{d}S \, n_i e^{ik_z z} \ . \tag{4.15}$$

Taking the surface $\Sigma$ to be the $xy$ plane, (4.15) evaluates to $\mathcal{Q} \propto \rho_{zz\cdots z}$, implying that $\rho_{zz\cdots z}$ is unable to decay in time. On the other hand, taking $\Sigma$ to be the $yz$ or $zx$ planes places no constraints on the components $\rho_{xz\cdots z}$ and $\rho_{yz\cdots z}$, since (4.15) evaluates to $\mathcal{Q} = 0$. Furthermore, the one-form symmetry does not constrain any components of $\rho$ orthogonal to the projector $\hat{k}_{i_2} \cdots \hat{k}_{i_n}$, where $\hat{\boldsymbol{k}}$ is the unit vector in the direction of $\boldsymbol{k}$.

### 4.5. Conditions leading to particular higher-rank theories

This origin of the one-form symmetry may alternatively be seen by taking $n - 1$ derivatives of the continuity equation, defining $\widetilde{J}_{i_1} \equiv \frac{1}{n} \partial_{i_2} \cdots \partial_{i_n} J_{i_1,i_2,\dots,i_n}$

$$\partial_t \widetilde{\rho}_i + \epsilon_{ik\ell} \partial_k \widetilde{J}_\ell = 0 \,. \tag{4.16}$$

Hence, common to all systems obeying generalized Maxwell's equations of the form (4.2) is a one-form symmetry of the conserved density $\widetilde{\rho}_{i_1} \equiv \partial_{i_2} \cdots \partial_{i_n} \rho_{i_1,\dots,i_n}$. In order to proceed in the reverse direction, i.e., from the equation for a one-form symmetry (4.16) to the continuity equation (4.10) for the object $\rho$ that transforms in the $(2n+1)$-dimensional irrep of $\mathsf{SO}(3)$, we must impose supplementary constraints on $\widetilde{\rho}_i$. First,

$$\int \mathrm{d}^3 x \, f \widetilde{\rho}_i = 0 \qquad \text{where} \qquad \partial_{i_1} \cdots \partial_{i_{n-1}} f = 0 \,, \tag{4.17}$$

for sufficiently well behaved $\widetilde{\rho}_i$. That is, multipole moments up to and including order $n - 2$ must strictly vanish (for the rank-two case (3.23), this reduces to total charge, while for the rank-one case there are no supplementary constraints on $\widetilde{\rho}_i$). Meanwhile, the tracelessness condition on $\rho_{i_1,i_2,\dots,i_n}$

---

[11] The terms included in Eq. (4.13) are sufficient to remove the trace part for $n < 6$. Including the second term only is sufficient for $n < 4$.

maps to

$$\int \mathrm{d}^3 x \, \widetilde{\rho}_i x_i (x_{i_3} \cdots x_{i_n}) = 0 \,. \tag{4.18}$$

The above represents $\binom{d+n-3}{n-2}$ independent constraints, which equals the number of independent components in the trace. The constraints that enforce symmetry, on the other hand, are given by

$$\int \mathrm{d}^3 x \, \epsilon_{kj\ell} \widetilde{\rho}_j x_\ell (x_{i_3} \cdots x_{i_n}) = 0 \,. \tag{4.19}$$

That the constraints (4.17) to (4.19) are sufficient to "canonically" determine the rank-$n$ continuity equation can be argued as follows. First, note that we can always write (nonuniquely) $\widetilde{\rho}_{i_1} = \partial_{i_2} \cdots \partial_{i_n} \rho_{i_1,\dots,i_n}$. The constraints on the various moments of $\widetilde{\rho}_i$ in (4.17) can be satisfied by introducing the higher rank object $\rho_{i_1,\dots,i_n}$ that is well-behaved at infinity by direct analogy with the arguments presented for the rank-two case in Sec. 3.3. Next, we make use of the symmetry constraints (4.19). In writing $\widetilde{\rho}_{i_1} = \partial_{i_2} \cdots \partial_{i_n} \rho_{i_1,\dots,i_n}$, we can assume that $\rho_{i_1,i_2,\dots,i_n} = \rho_{i_1,(i_2,\dots,i_n)}$ due to the commutativity of derivatives. Integrating (4.19) by parts $n-1$ times gives us that $\rho_{j(k,i_3,\dots,i_n)} = \rho_{k(j,i_3,\dots,i_n)}$, which implies that the tensor is *fully* symmetric, up to higher derivative corrections, the possibility of which we ignore. Tracelessness of $\rho_{i_1,\dots,i_n}$ then follows from integrating (4.18) by parts $n-1$ times, i.e., $\rho_{i,i,i_3,\dots,i_n} = 0$. Akin to the manipulations in Eq. (3.29), the corresponding constraints placed on the current $\widetilde{J}_i$ are found by taking the time derivative of (4.19), the precise details of which are deferred to Appendix B. There, we discuss carefully the full reconstruction of the continuity equation (4.10) in the specific setting of a rank-three theory. The key steps are as follows: (i) the constraints on $\widetilde{J}_i$ motivate the introduction of the rank-$n$ current $J_{I_n}$; (ii) symmetry of $\rho_{I_n}$ restricts the continuity equation to be of the form (4.10), but $J_{I_n}$ needn't be symmetric or traceless; (iii) tracelessness of $\rho_{I_n}$ then restricts $J_{I_n}$ to be fully symmetric and traceless.

## 5. MAGNETIC SUBDIFFUSION

We now consider a higher-rank theory that exhibits *subdiffusion* of magnetic field lines, the "traceful vector charge theory" of Ref. [32], in which the electric and magnetic charge (monopole) densities are vector valued.

### 5.1. Maxwell's equations with vector charge

The rank-two Maxwell's equations for symmetric tensor $E$ and $B$ fields and vector densities for the matter content are given by

$$\partial_j E_{ij} = \frac{1}{\varepsilon}\rho_i^{(e)} \tag{5.1a}$$

$$\partial_j B_{ij} = \mu\,\rho_i^{(m)} \tag{5.1b}$$

$$\partial_t B_{ij} = \epsilon_{ik\ell}\epsilon_{jmn}\partial_k\partial_m E_{\ell n} - \mu\,J_{ij}^{(m)} \tag{5.1c}$$

$$\partial_t E_{ij} = -\frac{1}{\mu\varepsilon}\epsilon_{ik\ell}\epsilon_{jmn}\partial_k\partial_m B_{\ell n} - \frac{1}{\varepsilon}J_{ij}^{(e)}\,. \tag{5.1d}$$

Unlike sections 3 and 4, the tensor fields $E_{ij}$ and $B_{ij}$ now transform in a reducible representation of $\mathsf{SO}(3)$, $\mathbf{5}\oplus\mathbf{1}$. Continuity equations for the electric and magnetic charges can be recovered by taking the divergence on one index of Faraday's (5.1c) and Ampère's (5.1d) laws, respectively. The continuity equations are given by

$$\partial_t\,\rho_i^{(e/m)} + \partial_j J_{ij}^{(e/m)} = 0\,. \tag{5.2}$$

### 5.2. Hydrodynamic interpretation

Like the traceless scalar charge theory and rank-one electromagnetism, Maxwell's equations can be interpreted as continuity equations for the rank-two electric and magnetic fields, $E_{ij}$ and $B_{ij}$, in the absence of their corresponding matter.

We begin by considering the matter-free limit. In the absence of electric and magnetic matter, both $E_{ij}$ and $B_{ij}$ are conserved densities. The fields obey wavelike equations, which derive straightforwardly using the same machinery employed in previous sections, and take the form

$$\partial_t^2 E_{ij} = \tilde{c}^2\left(-\partial_i\partial_j\partial_k\partial_m E_{km} + \partial^2\partial_k\partial_j E_{ki} + \partial^2\partial_k\partial_i E_{kj} - \partial^4 E_{ij}\right)\,, \tag{5.3}$$

for $E_{ij}$, and the equation of motion for $B_{ij}$ takes the same form due to electromagnetic duality. We have defined $\mu\,\varepsilon\,\tilde{c}^2 = 1$, although it should be noted that—in contrast to previous sections—$(\mu\varepsilon)^{-1/2}$ (and hence $\tilde{c}$) no longer has the dimensions of a speed.

For wavevector $\boldsymbol{k}$ oriented in the $\hat{\boldsymbol{z}}$ direction, the normal modes are given by

$$\omega = \tilde{c}\,k_z^2 \times \begin{cases} 0 & E_{zi} \\ 1 & E_{xx}, E_{yy}, E_{xy} \end{cases}, \tag{5.4}$$

corresponding to three quadratically dispersing modes. This is to be expected given that the symmetric tensor, $E_{ij}$, has six independent degrees of freedom, with three components removed by the Gauss's law constraints (5.1a) and (5.1b). In fact, Gauss's law constrains $k_z E_{zi} = 0$, which freezes the modes $E_{zi}$.

Next we consider how the hydrodynamic description is altered in the presence of electric charges (with all vector components). The effect of, say, electric matter is to break the conservation law

associated to $E_{ij}$ while preserving the conservation law associated to $B_{ij}$. Quasi-hydrodynamics dictates that the conservation law for $E_{ij}$ is should be broken in the most general manner permitted by symmetry constraints

$$\partial_t E_{ij} + \tilde{c}^2 \epsilon_{ik\ell}\epsilon_{jmn}\partial_k\partial_m B_{\ell n} = -\frac{1}{3\tau_1}\delta_{ij}E_{kk} - \frac{1}{\tau_5}\left(E_{ij} - \frac{1}{3}\delta_{ij}E_{kk}\right), \tag{5.5}$$

where $\tau_1$ and $\tau_5$ are phenomenological parameters characterizing the decay rate of the trace part and the traceless symmetric part of $E_{ij}$, respectively. The right-hand side of Eq. (5.5) represents the most general structure permitted by $\mathsf{SO}(3)$ rotational invariance; the fact that $E_{ij}$ transforms in a reducible representation of $\mathsf{SO}(3)$ implies that the "electrical condictivity" is no longer characterized by a single time scale in general (as was the case in all prior sections). Specifically, as noted above Eq. (3.6), $\mathsf{SO}(3)$ symmetry forces the electrical conductivity to be of the form $\sigma_{ijk\ell} = \alpha\delta_{ij}\delta_{k\ell} + \beta\delta_{ik}\delta_{j\ell} + \gamma\delta_{i\ell}\delta_{jk}$, which for $E_{ij}$ belonging to the reducible $\mathbf{5}\oplus\mathbf{1}$ representation gives $\varepsilon\tau_1^{-1} = 3\alpha$ and $\varepsilon\tau_5^{-1} = \beta + \gamma$. In the long time limit, $\tau_1\partial_t \ll 1$ and $\tau_5\partial_t \ll 1$, substituting (5.5) into (5.1c) gives

$$\partial_t B_{ij} = -\tau_5\tilde{c}^2\left(-\partial_i\partial_j\partial_k\partial_m B_{km} + \partial^2\partial_k\partial_j B_{ki} + \partial^2\partial_k\partial_i B_{kj} - \partial^4 B_{ij}\right)$$
$$+ \frac{1}{3}(\tau_5 - \tau_1)\tilde{c}^2\left(\delta_{ij}\partial^2 - \partial_i\partial_j\right)\left(\delta_{km}\partial^2 - \partial_k\partial_m\right)B_{km}, \tag{5.6}$$

and the quasi-normal modes for a wavevector, $\boldsymbol{k}$, oriented in the $\hat{\boldsymbol{z}}$ direction are

$$\omega = -\mathrm{i}\,\tilde{D}\,k_z^4 \times \begin{cases} 0 & B_{zj} \\ 1 & B_{xy}, B_{xx} = -B_{yy} \\ \frac{1}{3}\left(1 + \frac{2\tau_1}{\tau_5}\right) & B_{xx} = B_{yy} \end{cases}, \tag{5.7}$$

with $\tilde{D} = \tau_5\,\tilde{c}^2$. In the special case $\tau_1 = \tau_5$, the normal mode structure mirrors that of the matter-free case, but with subdiffusing—rather than propagating—modes. When the two time scales differ, $\tau_1 \neq \tau_5$, the quasinormal mode corresponding to the trace part of $B_{ij}$ decays with a different rate. In the presence of a Gauss law constraint, the three nondecaying modes, $B_{zj}$, are removed.

Note that in the presence of magnetic charge, the regime of validity of (5.7), i.e., the length and time scales over which magnetic subdiffusion occurs, is determined by analogy to previous sections, with modifications due to the higher-order nature of the hydrodynamic equations of motion.

### 5.3. One-form symmetries

The continuity equation for the rank-two magnetic field takes the general form

$$\partial_t\rho_{ij} + \partial_k\partial_m J_{ijkm} = 0, \tag{5.8}$$

where both the rank-two charge, $\rho_{ij}$, and rank-four current, $J_{ijkm} = \epsilon_{ik\ell}\epsilon_{jmn}J_{\ell n}$ transform as the reducible representation $\mathbf{1}\oplus\mathbf{5}$ of $\mathsf{SO}(3)$, corresponding to symmetric but not traceless rank-two

tensors. The conserved quantities associated to this continuity equation are of the form

$$\mathcal{Q}[f] \equiv \int_{\mathbb{R}^3} \mathrm{d}^3 x \, f_{ij} \rho_{ij} \tag{5.9}$$

where the symmetric tensor $f_{ij}$ satisfies

$$\epsilon_{ik\ell} \partial_k \epsilon_{jmn} \partial_m f_{ij} = 0 \,. \tag{5.10}$$

Note the similarity to (3.12), which has the curl acting only on a single tensor index of $f_{ij}$. Here, we obtain an infinite family of solutions of the form

$$f_{ij}(\boldsymbol{x}) = \partial_i \Psi_j + \partial_j \Psi_i \,, \tag{5.11}$$

where $\Psi_i(\boldsymbol{x})$ are arbitrary vector-valued functions. As in Sec. 3.3, the absence of additional solutions can be justified using the "scalar-vector-tensor" decomposition of $f_{ij}$. Recall that the tensor contribution, $f_{ij}^T \subset f_{ij}$, can be written in terms of the tensor $k^4 T_{ij}(\boldsymbol{k}) = -(\epsilon_{iab} k_a)(\epsilon_{jcd} k_c) f_{bd}$, up to the usual ambiguity in Helmholtz decompositions, which arises from additional contributions that are projected out when reconstructing $f_{ij}$ according to Eq. (3.17). Equation (5.10) therefore demands that $T_{ij}(\boldsymbol{k}) = 0$, leaving only the contributions from the "scalar" and "vector" terms, $f_{ij}^\perp$ and $f_{ij}^\parallel$, respectively. In momentum space, the general solution therefore assumes the form

$$f_{ij}(\boldsymbol{k}) = -k_i k_j \Phi + (\epsilon_{iab} k_a v_b) k_j + k_i (\epsilon_{jab} k_a v_b) \,, \tag{5.12}$$

parametrized in terms of the scalar $\Phi$ and the vector $v_i$. Note that the scalar term differs from (3.17) since $f_{ij}$ is not necessarily traceless. Equation (5.12) can be rewritten as

$$f_{ij}(\boldsymbol{k}) = \left[ \epsilon_{iab} k_a v_b - \tfrac{1}{2} k_i \Phi \right] k_j + k_i \left[ \epsilon_{jab} k_a v_b - \tfrac{1}{2} k_j \Phi \right] \,, \tag{5.13}$$

where the expression in the square brackets is identified as the Helmholtz decomposition of a vector $\Psi_i$, i.e., $\Psi_i = \epsilon_{iab} k_a v_b - \tfrac{1}{2} k_i \Phi$. We therefore recover the solution anticipated in Eq. (5.11), parametrized by the arbitrary vector field $\Psi_i(\boldsymbol{x})$.

To shed light on the conservation laws implied by the solution (5.11), we make use of the vector-valued indicator functions

$$\Psi_{i;k,V}(\boldsymbol{x}) = \delta_{ik} \times \begin{cases} 1 & \boldsymbol{x} \in V \,, \\ 0 & \boldsymbol{x} \notin V \,, \end{cases} \tag{5.14}$$

which lead to the three-component conserved quantity

$$\mathcal{Q}_i[S] = \int_S \mathrm{d}S \, \rho_{ij} \hat{n}_j \tag{5.15}$$

for each choice of surface $S = \partial V$. Since there are three conserved quantities associated with each surface $S$, the continuity equation effectively describes three one-form symmetries.

However, the three one-form symmetries are not completely independent, as the conserved

charges satisfy nontrivial constraints. Consider the conserved quantities

$$\mathcal{Q}_i(x_j) = \int \rho_{ij} \, \mathrm{d}x_k \mathrm{d}x_\ell \qquad \text{for} \ \ j \neq k \neq \ell. \tag{5.16}$$

These conserved quantities are a particular case of (5.15) where the surface, $S$, is taken to be the $x_k x_\ell$ plane at a given $x_j$. The constraint that $\mathcal{Q}_i(x_j)$ must satisfy is

$$\int \mathrm{d}x_j \, \mathcal{Q}_i(x_j) = \int \mathrm{d}x_i \, \mathcal{Q}_j(x_i), \tag{5.17}$$

where there is no sum on the repeated indices. The constraint (5.17) arises from the fact that the LHS is equal to $\int \mathrm{d}^3 x \, \rho_{ij}$ while the RHS is equal to $\int \mathrm{d}^3 x \, \rho_{ji}$. Equality follows from symmetry of the conserved density $\rho_{ij}$.

We now argue that the converse is true —that is, any theory hosting three independent one-form symmetries subject to the constraint (5.17) is necessarily the vector charge theory. We label our three conserved densities by $\rho_i^{(a)}$, where $a \in \{1, 2, 3\}$ is (for the moment) a flavor index labeling the conserved densities and $i$ is the usual spatial index. For each $a$, the charges satisfy the continuity equation for a one-form symmetry:

$$\partial_t \rho_i^{(a)} + \epsilon_{ik\ell} \partial_k J_\ell^{(a)} = 0. \tag{5.18}$$

To each plane perpendicular to a coordinate direction $x_i$ we associate three conserved charges

$$\mathcal{Q}^{(a)}(x_i) \equiv \int \rho_i^{(a)} \mathrm{d}x_j \mathrm{d}x_k \qquad \text{for} \ \ i \neq j \neq k. \tag{5.19}$$

labeled by the flavor index $a$. To these conserved charges we impose the constraint

$$\int \mathrm{d}x_i \, \mathcal{Q}^{(a)}(x_i) = \int \mathrm{d}x_a \, \mathcal{Q}^{(i)}(x_a), \tag{5.20}$$

which is the analogue of (5.17). Note that the constraint requires that the flavor index $a$ be identified with a spatial index that can be used to label the coordinates, so we will neglect the parentheses henceforth. Expanding and rearranging the constraint (5.20) yields

$$\int \mathrm{d}^3 x \, \left[\rho_i^a - \rho_a^i\right] = 0. \tag{5.21}$$

We conclude that the antisymmetric part of $\rho_i^a$ identically vanishes or is the divergence of a higher-rank object. For simplicity we assume the former; the latter reduces to the former at long wavelengths. There is hence no reason to distinguish between "raised" and "lowered" indices, and we rewrite $\rho_i^a \to \rho_{ai}$. The constraint that $\rho_{ai}$ is symmetric leads to a constraint on $J_{a\ell}$: It must be of the form $\epsilon_{amn}\partial_m J_{n\ell}$ so that the second term of (5.18) is symmetric. We then recover the continuity equation

$$\partial_t \rho_{ia} + (\epsilon_{ik\ell}\partial_k)(\epsilon_{amn}\partial_m)J_{\ell n} = 0. \tag{5.22}$$

This completes the understanding of the features of the subdiffusive normal modes in (5.7): The mode structure arises from the three one-form symmetries, while subdiffusion arises from the constraint (5.17), which forces a second derivative in the continuity equation for $\rho_{ij}$.

## 6.  CONCLUSION

We have presented a hydrodynamic formulation capable of dealing with *gauged* multipolar symmetries, such as are expected to arise in fracton phases of matter. The formulation we have presented is a natural generalization of the treatments of ordinary (Maxwell) magnetohydrodynamics based on higher-form symmetries. This (somewhat abstract) formulation has the advantage of not being limited in validity to the weak-coupling regime, unlike more semi-microscopic approaches where one simply couples a fracton fluid (as developed in, e.g., Ref. [14]) to a higher-rank gauge theory. Instead, we have argued that "fracton magnetohydrodynamics" is best understood in terms of *one-form* symmetries, just like conventional magnetohydrodynamics [48], and the hydrodynamic objects are (linelike) symmetry charges of this one-form symmetry, which may be viewed as "generalized magnetic flux lines."

One surprising feature is that the "higher rank" fractonic gauge theories generically exhibit *diffusion* of magnetic flux lines, in contrast to the subdiffusion of charge that is seen in theories with global multipolar symmetry. To gain intuition for this absence of subdiffusion, it is helpful to recall that whereas charge diffuses in a theory with a global $\mathsf{U}(1)$ symmetry, once the symmetry gets gauged, the charge relaxes exponentially, being driven by long range interactions carried by the gauge fields. Similarly, the "subdiffusion of charge" obtained in theories with global multipolar symmetry generically gives way to exponential relaxation when the symmetry is gauged. The hydrodynamic modes involve not the charges, but rather the *flux lines*, and these generically relax diffusively. Nevertheless, theories with subdiffusion of magnetic field lines can also be accessed, and we have provided a specific example thereof.

The theories we present describe the generic long-time description of the quantum dynamics of fractonic phases (at nonzero charge density) exhibiting *gauged*—as opposed to global—multipolar symmetries (as relevant to spin liquids and fracton phases). We develop a symmetry-based approach describing arbitrary higher-rank theories of this type, and showcase the subdiffusion of magnetic fields as an example of the exotic universal dynamics that may arise in this context.

One obvious direction for the future is to aim to apply this formalism to emerging experiments on quantum spin liquids, both conventional and fractonic. Any such program would need to be driven by experiment, so we do not discuss it further in this (theoretically focused) manuscript. However, there are also important conceptual points of principle that should be cleaned up in future work. For example, how can the effects of momentum conservation be incorporated into our hydrodynamic framework? Formally, this necessitates coupling the higher-rank gauge theory to curved spacetime, which was done for MHD in Ref. [48]. However, this leads to technical challenges associated to the fact that defining moments of charge on curved spacetime is difficult [72]. Additionally, we have restricted ourselves in this manuscript to systems in three spatial dimensions. Are there surprises if one changes the dimension? Furthermore, the theories we have herein considered involve gauged $\mathsf{U}(1)$ symmetries. However, going from such theories to the kinds of lattice models beloved of the

quantum information community requires a sequence of Higgs and partial confinement transitions [73]. What happens to the hydrodynamic theory as we go through these transitions? Or again: thus far we have considered gauge theories of *Abelian* fractons. What if we move to non-Abelian generalizations? It was argued in [74] that imposing non-Abelian multipolar global symmetries would totally trivialize the dynamics, but the same need not be true of *gauged* non-Abelian symmetries, and the magnetohydrodynamics of non-Abelian fractonic systems could be a particularly fruitful problem for future work, extending the literature on non-Abelian versions of magnetohydrodynamics [75–78].

## ACKNOWLEDGEMENTS

MQ was supported by the National Defense Science and Engineering Graduate Fellowship (NDSEG) program. AL was supported by the Gordon and Betty Moore Foundation's EPiQS Initiative via Grant GBMF10279, by the National Science Foundation via CAREER Grant DMR-2145544, and by a Research Fellowship from the Alfred P. Sloan Foundation under Grant FG-2020-13795. OH and RN were supported by the U.S. Department of Energy, Office of Science, Basic Energy Sciences, under Award #DE-SC0021346. AJF was supported in part by a Simons Investigator Award via Leo Radzihovsky.

## Appendix A: Charge and current belonging to different irreps

### A.1. Two-form symmetry: Irrep 1

Consider a theory that contains a vector charge $\rho_i$ belonging to the **3** of $\mathsf{SO}(3)$, but whose associated current $J_{ij}$ transforms instead in the **1**, i.e., the trivial or scalar irrep of $\mathsf{SO}(3)$. In this case the current may be parametrized in terms of the scalar $J$:

$$J_{ij} = J\,\delta_{ij}\,, \tag{A.1}$$

and the continuity equation for vector charge density becomes

$$\partial_t \rho_i + \partial_i J = 0\,. \tag{A.2}$$

As in the main text, we define the putatively conserved quantity $\mathcal{Q}[f_i]$ parametrized by vector fields $f_i(\boldsymbol{x})$, i.e., $\mathcal{Q}[f_i] \equiv \int \mathrm{d}^3x\, f_i \rho_i$. Using the continuity equation (A.2), we obtain the following constraints on the field $f_i$

$$\frac{\mathrm{d}}{\mathrm{d}t}\mathcal{Q}[f_i] = \int \mathrm{d}^3x\, f_i \partial_t \rho_i = -\int \mathrm{d}^3x\, f_i\, \partial_i\, J = \int \mathrm{d}^3x\, J\, \partial_i f_i\,, \tag{A.3}$$

where in the final equality we have integrated by parts. We then find that, so long as $f_i$ is divergence-free,

$$\partial_i f_i = 0\,, \tag{A.4}$$

we are guaranteed that $\mathcal{Q}[f_i]$ is a conserved quantity of the theory.

As is well known, there are infinitely many solutions to (A.4), which may be expressed by writing $f_i = \epsilon_{ijk}\Omega_{jk}$ using an antisymmetric tensor $\Omega_{jk} = -\Omega_{kj}$, in which case (A.4) amounts to the statement that the two-form, $\Omega$, is "closed," i.e., $d\Omega = 0$. Up to topological effects (which we do not consider here), this implies that $\Omega = d\alpha$ is exact, or that

$$f_i = \epsilon_{ijk}\partial_j\alpha_k \, , \tag{A.5}$$

for any function $\alpha_k$ (i.e., the vector field $f_i$ is the curl of some vector-valued function).

In a similar manner to the indicator functions chosen to elucidate the conservation of flux through surfaces in, e.g., Sec. 2.3 of the main text, here it is instructive to choose a particular form for the functions $\alpha_k(\boldsymbol{x})$. Let $\gamma$ denote a curve (closed, or infinite in extent) in $\mathbb{R}^3$, and consider

$$\alpha_k(\boldsymbol{x}) \equiv \int_\gamma \frac{\epsilon_{ijk}\,dy_i\,(y_j - x_j)}{|\boldsymbol{y} - \boldsymbol{x}|^3} \, , \tag{A.6}$$

where $\boldsymbol{y}$ denotes the line integral along $\gamma$ and $\boldsymbol{x}$ denotes an arbitrary point in $\mathbb{R}^3$. Using the textbook Biot–Savart law, we see that

$$f_i(\boldsymbol{x}) = \int_\gamma dy\,\delta^3\,(\boldsymbol{x} - \boldsymbol{y})\,n_i(\boldsymbol{y}) \, , \tag{A.7}$$

where $\boldsymbol{n}(\boldsymbol{y})$ is the unit vector along $\gamma$ at point $\boldsymbol{y}$. We can write this more elegantly: The conserved charges are generated by the different curves $\gamma$ and using the notation that $\rho$ is a one-form,

$$\mathcal{Q}_\gamma = \int_\gamma \rho \, , \tag{A.8}$$

is a conserved quantity. For each curve $\gamma$ there exists a corresponding conserved charge, $\mathcal{Q}_\gamma$.

### A.2.  Scale- and rotation-invariant hydrodynamics: Irrep 5

Next, suppose $J_{ij}$ transforms in the **5** of $\mathsf{SO}(3)$, corresponding to traceless symmetric tensors, which may be parametrized by explicitly removing the antisymmetric and trace parts of a generic rank-two current tensor

$$\partial_t\rho_i + \left[\partial_j J_{ij} + \partial_j J_{ji} - \frac{2}{3}\partial_i J_{jj}\right] = 0 \, , \tag{A.9}$$

where the term in the square brackets is manifestly symmetric and traceless. Looking for conserved quantities $\mathcal{Q}[f_i]$ parametrized by vector-valued functions $f_i(\boldsymbol{x})$ and repeating the same logic as before, we find that

$$\partial_i f_j + \partial_j f_i - \frac{2}{3}\delta_{ij}\partial_k f_k = 0 \tag{A.10}$$

in order for $\mathcal{Q}[f_i]$ to be conserved. In three spatial dimensions, there is a finite list of solutions to (A.10), given explicitly by

$$f_i = \alpha \, x_i + \epsilon_{ijk} \, x_j \beta_k \,, \tag{A.11}$$

with $\alpha$ and $\beta_k$ scalar and vector constants, respectively. Hence, (A.9) will generally only lead to seven conserved quantities (four additional conservation laws from (A.11), and the three original charges corresponding to $f_i = 1$, since $\rho_i$ is a conserved density). If we were to interpret $\rho_i$ as a velocity field, then the conserved quantities would correspond to momentum ($\rho_i$), angular momentum ($\epsilon_{ijk} \, x_j \rho_k$), and "dilatation" ($x_i \rho_i$).

### A.3. Mixing and matching

In general, it's possible that that the currents may not transform as a single irrep but instead as a direct sum of different irreps. As an example, hydrodynamics with rotation invariance but *without* scale invariance has a current $J_{ij}$ that transforms in the $\mathbf{1} \oplus \mathbf{5}$ representation. In this case, the current decomposes as

$$J_{ij} = J \, \delta_{ij} + \widetilde{J}_{ij} \,, \tag{A.12}$$

where $J$ and $\widetilde{J}_{ij}$ transform in the $\mathbf{1}$ and $\mathbf{5}$ irreps, respectively. The continuity equation then reads

$$\partial_t \rho_i + \left[ \partial_j J_{ij} + \partial_j J_{ji} - \frac{2}{3} \partial_i J_{jj} \right] + \frac{1}{3} \partial_i J_{jj} = 0 \,. \tag{A.13}$$

The quantity $\mathcal{Q}[f_i]$ is conserved when both (A.4) *and* (A.10) are satisfied simultaneously. The finite list of solutions in (A.11) is reduced to

$$f_i = \epsilon_{ijk} \, x_j \, \beta_k \tag{A.14}$$

so that the "dilatation" $x_i \rho_i$ is no longer a conserved quantity. This is an example of the general situation wherein the current decomposes into irreducible representations; if there exists a list of conserved quantities associated to each irrep, then the set of conserved quantities is reduced to the intersection of these lists when the current has nonzero overlap with multiple irreps.

### Appendix B: From constraints to the rank-$n$ continuity equation

In this appendix we explicitly work through the steps that one may take to "canonically" derive the rank-three continuity equation from the associated constraints on the density $\widetilde{\rho}_i$, which obeys the equation of motion

$$\partial_t \widetilde{\rho}_i + \epsilon_{ik\ell} \partial_k \widetilde{J}_\ell = 0 \tag{4.16}$$

associated with a one-form symmetry. The generalization to higher rank theories, $n > 3$, follows an identical line of reasoning.

In Sec. 4.5, we described how the constraints (4.17) to (4.19) lead one to consider a symmetric, traceless rank-$n$ tensor $\rho_{i_1,\dots,i_n}$, related to $\widetilde{\rho}_i$ via $\widetilde{\rho}_{i_1} = \partial_{i_2} \cdots \partial_{i_n} \rho_{i_1,\dots,i_n}$, that is well behaved at infinity. We now proceed by considering the constraints placed on the effective current $\widetilde{J}_i$ associated with the density $\widetilde{\rho}_i$. Specializing to rank-three and taking the time derivative of Eq. (4.19), we find

$$\frac{\mathrm{d}}{\mathrm{d}t} \int \mathrm{d}^3x \, \epsilon_{kj\ell} \widetilde{\rho}_j x_\ell x_i = - \int \mathrm{d}^3x \, \epsilon_{kj\ell} \epsilon_{jmn} x_\ell x_i \partial_m \widetilde{J}_n = 0 \,. \tag{B.1}$$

Consequently, the current $\widetilde{J}_i$ is also constrained, and it is more natural to write $\widetilde{J}_i = \frac{1}{3} \partial_j \partial_k J_{ijk}$ in terms of a rank-three tensor that need only vanish at infinity. To see this, we write (B.1) in terms of the new tensor $J_{ijk}$ and integrate by parts:

$$0 = 3 \int \mathrm{d}^3x \, \epsilon_{kj\ell} \epsilon_{jmn} x_\ell x_i \partial_m \widetilde{J}_n = \int \mathrm{d}^3x \, \epsilon_{kj\ell} \epsilon_{jmn} x_\ell x_i \partial_m \partial_a \partial_b J_{nab} = \int \mathrm{d}^3x \, \epsilon_{kj\ell} \epsilon_{jmn} \partial_m J_{n\ell i} \quad \text{(B.2)}$$

where we have used the fact that $J_{ijk}$ is symmetric in [at least] its last two indices. Contracting the two Levi-Cevita symbols, we arrive at

$$\int \mathrm{d}^3x \left( \partial_\ell J_{k\ell i} - \partial_k J_{\ell\ell i} \right) = 0 \,. \tag{B.3}$$

If the second term vanishes, then (B.3) implies that $J_{ijk}$ that decay away sufficiently quickly as $|\boldsymbol{x}| \to \infty$ will satisfy the constraint in Eq. (B.1). That the second term should vanish (since $J_{ijk}$ should be symmetric and traceless) is not immediately apparent; we show that this must be the case below. Writing (4.16) in terms of the new rank-three degrees of freedom

$$\partial_j \partial_k \left( \partial_t \rho_{ijk} + \frac{1}{3} \epsilon_{imn} \partial_m J_{njk} \right) = 0 \,. \tag{B.4}$$

To derive an equation of motion for the higher rank object $\rho_{ijk}$, we integrate up Eq. (B.4), leading to

$$\partial_t \rho_{ijk} + \frac{1}{3} \left( \epsilon_{imn} \partial_m J_{njk} + \epsilon_{jmn} \partial_m \Lambda_{nki} + \epsilon_{kmn} \partial_m \Lambda'_{nij} \right) = 0 \,, \tag{B.5}$$

where last two terms on the right are the most general terms annihilated by the derivatives $\partial_j \partial_k$ compatible with the index structure in Eq. (B.4). We now require that the equations must respect the symmetry and tracelessness of $\rho$, which imposes stringent constraints on the permitted form of $\Lambda$ and $\Lambda'$. First, requiring that $\rho_{ijk} = \rho_{jik}$, we find that

$$\epsilon_{imn} \partial_m (J_{njk} - \Lambda_{nkj}) + \epsilon_{jmn} \partial_m (\Lambda_{nki} - J_{nik}) + \epsilon_{kmn} \partial_m (\Lambda'_{nij} - \Lambda'_{nji}) \stackrel{!}{=} 0 \,. \tag{B.6}$$

The final term on the left-hand side suggests that $\Lambda'_{nij} = \Lambda'_{nji}$, up to terms that contain a higher number of derivatives. A similar argument can be made for the other "integration constant", $\Lambda$, by requiring that $\rho_{ijk} = \rho_{kji}$. We therefore take both $\Lambda$ and $\Lambda'$ to be symmetric in their last two

indices. Under this assumption, the first two terms in (B.6) can be satisfied, to lowest order in derivatives, by $\Lambda_{ijk} = J_{ijk}$. Similarly, $\Lambda'_{ijk} = J_{ijk}$ follows from symmetry of $\rho_{ijk}$ in its first and last indices. Requiring that $\rho$ remains traceless under time evolution leads to the requirement that

$$- \partial_t \rho_{iik} = \frac{1}{3} \left( 2\epsilon_{imn}\partial_m J_{nik} + \epsilon_{kmn}\partial_m J_{nii} \right) \overset{!}{=} 0 \,, \tag{B.7}$$

suggesting that we should take $J_{ijk}$ to be symmetric in its first two indices (making it fully symmetric when twinned with symmetry in its last two indices), *and* traceless. While a fully symmetric and traceless $J$ certainly satisfies (B.7), it turns out that this is not the only choice. There exists one further solution in which the current transforms in the reducible $\mathbf{3} \oplus \mathbf{3}$ representation:

$$J_{ijk} = \frac{2}{3}\left[\delta_{ij}\lambda_k + \delta_{ki}\lambda_j + \delta_{jk}\lambda_i\right] + \left[\frac{1}{3}\left(\delta_{ij}\lambda_k + \delta_{ik}\lambda_j\right) - \frac{2}{3}\delta_{jk}\lambda_i\right] = \delta_{ij}\lambda_k + \delta_{ik}\lambda_j \,, \tag{B.8}$$

parametrized by the vector field $\lambda_i(\boldsymbol{x})$. While the existence of such a solution may appear to imply that the rank-three continuity equation cannot be obtained from (4.16) and the corresponding constraints alone, we note that

$$\epsilon_{imn}\partial_m(\delta_{nj}\lambda_k + \delta_{nk}\lambda_j) + \epsilon_{jmn}\partial_m(\delta_{ni}\lambda_k + \delta_{nk}\lambda_i) + \epsilon_{kmn}\partial_m(\delta_{ni}\lambda_j + \delta_{nj}\lambda_i) = 0 \,, \tag{B.9}$$

i.e., the solution (B.8) does not contribute to the equation of motion for $\rho_{ijk}$ [as was the case for the trace part of $J_{ij}$ below Eq. (3.32)], and can therefore be disregarded. This leaves us with the equation of motion

$$\partial_t \rho_{ijk} + \frac{1}{3}\left( \epsilon_{imn}\partial_m J_{njk} + \epsilon_{jmn}\partial_m J_{nki} + \epsilon_{kmn}\partial_m J_{nij} \right) = 0 \,, \tag{B.10}$$

where both $\rho_{ijk}$ and $J_{ijk}$ transform in the $\mathbf{7}$ of $\mathsf{SO}(3)$, i.e., they are symmetric traceless rank-three tensors. This is precisely the form of the continuity equation in Eq. (4.10) of the main text.

---

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
