# Peer review of "Fracton magnetohydrodynamics"

_SciPost Physics_

## Round 1 · Referee Report · Anonymous (Referee 1) · 2022-6-15

Strengths

The paper is detailed and easy to follow.
Research topics are exciting and timely.

Weaknesses

No proposed realistic systems realize the interesting physics discussed in the manuscript.

Section 2 and the results of the matter-free limit in other sections are known (some of them are presented in Ref[32]).

Report

The authors of the manuscript offered the hydrodynamics description of the systems with higher rank gauge symmetry. The results are interesting and timely, given that fracton systems' dynamics have recently attracted attention from high-energy and condensed matter communities. The paper is scientifically sound and well written.

In conclusion, I recommend the paper for publication after the authors clarify my following questions and concerns

1) Could the author explicitly define the hydrodynamics regime for each case? What is the limit where the hydrodynamics formalism is valid? As my understanding, in graphene, the hydrodynamics regime is where the conserved momentum scattering dominates the momentum relaxation scattering.

2) From the divergent free condition of the magnetic field $\partial_i B_i=0$, by the Gauss theorem, the total magnetic flux through the boundary of any volume vanishes identically rather than conserved as in equation (2.13). Is this correct?

3) In the symmetric rank 2, the “electric conductivity” generally is a 4-indices tensor. What are the argument of the relation $J^{(e)}{ij}=-\frac{1}{\tau}E$ used in equation (3.6) and (5.5). Why the Ohmic conductivities of scalar charge and vector charge are the same? What are the assumed scattering mechanism that lead to this conclusion? Is the proposed “electric conductivity” unique with the assumption of rotational symmetry?

4) In the discussion of one-form symmetries for the conserved magnetic field in equations (3.10) and (5.8), the assumption that there is no magnetic charge was used implicitly. What is the argument for this assumption?

5) It looks like the energy of the photon in the matter-free limit for the traceless symmetric rank $n$ is given by $\omega=\frac{c k_z m}{n}$, is this correct?

6) The explicit forms of functions $f_i$, $f_{ij}$ in sections 2,3,5 remind me the gauge transformations of gauge fields $A_i$ and $A_{ij}$ with time independent gauge transformation. $A_i \to A_i + \partial_i \varphi$ for the usual $U(1)$ gauge theory, $A_{ij} \to A_{ij}+\partial_{i}\partial_j \Phi -\frac{1}{3} \delta_{ij} \partial^2 \Phi$ for scalar charge traceless symmetric tensor gauge theory, and $A_{ij} \to A_{ij} + \partial_i \Psi_j + \partial_j \Psi_i$ for vector charge tensor gauge theory. Is this just a coincident or they are related?

Minors:

1)The magnetic conductivity $\sigma_m$ wasn’t defined before used in equation (2.16) 2) Are $\tilde{J}_k$ in (2.21) and $J_k$ in (2.22) the same? 3) The diffusion constant at rank $n$ given by $D_m=\tau c^2 m^2/n^2$ only works for the traceless symmetric tensor case. It should be clarified explicitly in the introduction section.

Requested changes

See the report

  • validity: high
  • significance: good
  • originality: good
  • clarity: high
  • formatting: excellent
  • grammar: excellent

Author:  Oliver Hart  on 2022-08-16  [id 2733]

(in reply to Report 1 on 2022-06-15)

We thank the referee for reviewing our manuscript and for their favorable assessment. We provide a detailed point-by-point response below. A complete list of changes will accompany our resubmission.

1) Could the author explicitly define the hydrodynamics regime for each case? What is the limit where the hydrodynamics formalism is valid? As my understanding, in graphene, the hydrodynamics regime is where the conserved momentum scattering dominates the momentum relaxation scattering.

In graphene the term "hydrodynamic regime" is a bit of a misnomer: this regime is more accurately described as the "viscous regime". Both the "Ohmic" and "viscous" regimes can be thought of as hydrodynamic regimes, and correspond to different exact/approximate conservation laws. Which regime one is in depends on particular ratios of important length scales. In Ohmic flow, impurity scattering dominates, thereby relaxing momentum; in viscous flow, momentum-conserving electron-electron collisions dominate. By contrast, the "ballistic regime" is dominated by the channel's narrowness, and is not hydrodynamic, as interactions do not play a dominant role in the dynamics. In the context of graphene, all of these regimes refer to the flow of matter (electrons), while in our manuscript we consider the hydrodynamic relaxation of fields. Essentially, we use the term "hydrodynamics" more generally to refer to the coarse-grained description of relaxation to global equilibrium in systems with conserved charges. Finally, while the label "Ohmic" could be applied to many of the regimes we study—in the sense that the hydrodynamic mode has dispersion $\omega \sim -\mathrm{i}k^2$—we think this label would be misleading, because the "Ohmic" transport coefficient would not be proportional to the coefficient $\sigma$ in the quasihydrodynamic description with both $E$ and $B$ fields. We will elucidate this point further in response to point 1. of the third referee.

2) From the divergent free condition of the magnetic field $\partial_i B_i = 0$, by the Gauss theorem, the total magnetic flux through the boundary of any volume vanishes identically rather than conserved as in equation (2.13). Is this correct?

The referee is correct that the magnetic Gauss's law (2.1b) implies that the magnetic flux through any closed surface vanishes, as shown directly in (2.11). This is entirely compatible with (2.13), which derives from (2.1b). We present the latter equation—and Sec. 2.2.3 more generally—to give some intuition for one-form symmetries in Maxwellian theories, and to highlight connections between Gauss's law and Faraday's law.

3) In the symmetric rank 2, the "electric conductivity" generally is a 4-indices tensor. What are the argument of the relation $J_{ij}^{(\text{e})} = -\frac{1}{\tau} E_{ij}$ used in equation (3.6) and (5.5). Why the Ohmic conductivities of scalar charge and vector charge are the same? What are the assumed scattering mechanism that lead to this conclusion? Is the proposed "electric conductivity" unique with the assumption of rotational symmetry?

The justification, in general, for the relation $J_{ij}^{(\text{e})} \propto E_{ij}$ (such that the "electrical conductivity" reduces to a single scalar parameter) is the SO(3) symmetry assumed throughout the manuscript. When $E_{ij}$ transforms in an irreducible representation of SO(3), $J_{ij}^{(\text{e})} = \sigma_e E_{ij}$ is the most general form of Ohm's law permitted by symmetry. In Sec. 5, corresponding to the traceful vector charge theory, $E_{ij}$ transforms in a reducible representation, permitting additional structure in Ohm's law. In the updated manuscript, we have highlighted that the $\mathbf{1}$ and the $\mathbf{5}$ irreps can, in general, have differing relaxation rates, and we have explored the consequences thereof on the quasinormal mode structure. We thank the referee for drawing this point to our attention.

One of the advantages of our symmetry-based approach is that no scattering mechanisms need to be assumed in order to reach the above conclusions: We allow for the most general form of Ohm's law allowed by symmetry. In sections 2, 3, and 4, the conductivity $\sigma$ reduces to a single "phenomenological parameter".

4) In the discussion of one-form symmetries for the conserved magnetic field in equations (3.10) and (5.8), the assumption that there is no magnetic charge was used implicitly. What is the argument for this assumption?

We emphasize that the vanishing of magnetic charge density is only ever assumed to hold approximately. Since the theories that we discuss are ostensibly emergent, we cannot rule out the presence of magnetic charge. The limit of vanishing magnetic charge density, where the magnetic one-form symmetry is exact, is taken in order to describe an intermediate time regime that exhibits interesting hydrodynamic phenomena—this is worth studying independent of any proposed experimental realization. As we explain in sections 2.2.5 and 3.2.3, if magnetic charges are present, then our results apply to intermediate length and time scales. More precisely, if the magnetic one-form symmetry is broken with a time scale $\tau_m > \tau_e$ (in addition to the electric one-form symmetry being broken), then all fields decay exponentially (i.e., no diffusion) at times $t \gtrsim \tau_m$. To observe the "magnetic diffusion" regime in a realistic physical system, one therefore only requires a separation of scales between the decay times $\tau_e$ and $\tau_m$. This separation of scales is analogous to how one sees viscous flow in graphene: Momentum is not exactly conserved, but there is a parametric separation between the time scales of momentum relaxation and the relaxation of more generic ballistic modes.

5) It looks like the energy of the photon in the matter-free limit for the traceless symmetric rank $n$ is given by $\frac{ck_z m}{n}$, is this correct?

Yes. We have made this fact more explicit in the manuscript.

6) The explicit forms of functions $f_i$, $f_{ij}$ in sections 2,3,5 remind me the gauge transformations of gauge fields $A_i$ and $A_{ij}$ with time independent gauge transformation. $A_i \to A_i + \partial_i \varphi$ for the usual $U(1)$ gauge theory, $A_{ij} \to A_{ij} + \partial_i \partial_j \Phi -\frac13 \delta_{ij} \partial^2 \Phi$ for scalar charge traceless symmetric tensor gauge theory, and $A_{ij} \to A_{ij} + \partial_i \Psi_j + \partial_j \Psi_i$ for vector charge tensor gauge theory. Is this just a coincident or they are related?

For all cases considered in the manuscript, both the permitted gauge transformations of the vector potential $A$ and the functional forms of the conserved moments of charge density $f(\mathbf{r})$ coincide. In fact, it is the self-duality of Maxwell's equations that guarantees this feature (specifically under interchanging $E$ and $B$, up to factors of $c$). Time-independent gauge transformations are generated by the electric Gauss's law, since $E$ and $A$ are canonically conjugate to one another. The functional form of the conserved moments $f$, on the other hand, derives from Faraday's law (assuming that it is the magnetic charge density that [approximately] vanishes), giving rise to conservation of magnetic flux through closed surfaces (in the rank-one case). The quantities that are conserved via Faraday's law also follow from the magnetic Gauss law. In self-dual theories, the electric and magnetic Gauss's laws have the same form. In higher-rank theories that are not self-dual, such as the traceful scalar charge theory, the equivalence between allowed gauge transformations and the functional form of conserved moments breaks down: In this theory, the gauge transformations are of the form $A_{ij} \to A_{ij} + \partial_i \partial_j \alpha$ for some scalar function $\alpha$, while the conserved moments of the magnetic field tensor $B_{ij}$ are parametrized instead by functions $f_{ij} = \partial_i \Psi_j$ (since $B_{ij}$ isn't symmetric, there is no requirement that the tensor $f_{ij}(\mathbf{r})$ be symmetric either).

1) The magnetic conductivity $\sigma_m$ wasn't defined before used in equation (2.16)

We thank the referee for spotting this omission. We have modified the first sentence of Sec. 2.2.5 to include a definition of $\sigma_m$.

2) Are $\tilde{J}_k$ in (2.21) and $J_k$ in (2.22) the same?

Yes. We have removed the tilde from $\tilde{J}_k$ in (2.21) and in the subsequent paragraph to clarify this.

3) The diffusion constant at rank $n$ given by $D_m = \tau c^2 m^2 /n^2$ only works for the traceless symmetric tensor case. It should be clarified explicitly in the introduction section.

The referee is correct. We have modified the relevant sentence in the introduction.

---

## Round 1 · Referee Report · Anonymous (Referee 2) · 2022-6-27

Report

The authors develop a theory of magnetohydrodynamics for the higher-rank gauge fields arising in fractonic systems. The core results of the paper are both timely and interesting, for example, the striking prediction of magnetic subdiffusion in fractonic MHD.

However, I did find some aspects of the presentation distracting, particularly the repeated claim that the approach in this work is fundamentally more rigorous and well-controlled than standard, “phenomenological” magnetohydrodynamics. I agree that the idea of Ref. 48, that flux conservation in MHD can be thought of as a slow mode arising from higher-form symmetry, is an appealing way to interpret the MHD equations. But it seems to me that the actual physical predictions in the paper could have been derived entirely within the phenomenological framework that the authors suggest is obsolete.

I therefore request that the authors either globally soften their language before publication, or better justify their claims.

In particular, if there is anything in these results that could not have been derived using the standard method, it would help to make clear where the conclusions of the two approaches differ. If there is no material difference, it would be worth trying to address why the standard method works so well beyond weak coupling.

For example, the discussion in the paper (and the claim in the earlier Ref. 48) has me wondering whether the standard derivation of MHD needs to invoke a weak-coupling assumption at all. Supposing the system has even one hydrodynamic mode (e.g. energy), one can coarse-grain the Maxwell equations to obtain a system of macroscopic Maxwell equations at the length and time scales associated with this hydrodynamic mode (using one of the many possible approaches in the literature). The linearization of these equations then yields linear response dynamics of the full system of matter+fields, and as usual, will predict conservation of magnetic flux density/decay of the other fields.

This procedure works because the Maxwell equations are linear: the linearized, macroscopic Maxwell equations have the same mathematical form as the microscopic Maxwell equations. I wonder whether it is this aspect of the Maxwell theory (i.e. linearity) that means the phenomenological approach yields qualitatively correct predictions, rather than the weak-coupling assumption. (This might be one distinction from e.g. Yang-Mills.)

My specific comments on the manuscript are as follows:

1. On the question of standard MHD vs 1-form symmetries, please consider revising the following:
Abstract: “In contrast to phenomenological MHD, our approach based on higher-form symmetries provides a systematic treatment of hydrodynamics beyond weak coupling, and provides an effective hydrodynamic description, e.g., of quantum spin liquids, whose emergent gauge fields are strongly coupled.”

I think this sentence overstates the level of rigor of the paper, as it is written. The approach of the authors is also phenomenological and e.g. uses the relaxation time approximation. It is also only “systematic” insofar as the one-form symmetry is exact, which is never true for emergent gauge fields as arise in condensed matter settings.

p2: “Somewhat surprisingly, this was not carefully done even for ordinary electromagnetism until the past decade”

This seems to dismiss all the (vast) previous literature on this topic as “careless”, which is unlikely to be the case. In particular, whether the magnetic slow mode is interpreted as local conservation of flux density or as a consequence of global 1-form symmetry is to some extent a matter of taste, and does not obviously depend on the coupling strength. Please consider revising.

p4: “This approach, based on higher-form symmetries, has significant conceptual advantages over more familiar phenomenological derivations of MHD. Specifically, whereas phenomenological derivations of MHD are limited in their validity to regimes wherein the gauge fields and matter are weakly coupled [48], the symmetry-based approach involving higher forms is valid for arbitrary coupling. In the context of emergent gauge theories in quantum spin liquids, there is no reason a priori to expect weak coupling between the matter and gauge fields [54]. An approach predicated on weak coupling is therefore suspect, and a more general approach—e.g., based on higher-form symmetries—is required…We emphasize that, while we focus on fractonic systems, our approach also applies to conventional spin liquids in the strongly coupled regime, where conventional phenomenological approaches may be uncontrolled.”

I am not entirely convinced by this line of reasoning for the reasons discussed above. First, the higher-form symmetry is itself usually an approximation in physical realizations where alpha is believed to be large and the gauge field is emergent. Second, it appears that the part of the paper relevant to conventional spin liquids (i.e. Sec. II) recovers the conclusions of the conventional MHD approach, as taken in Ref. [62].

I therefore ask that the authors either soften their wording, or more carefully justify how leading-order, linear-response MHD requires the weak-coupling assumption – if this recovers the same conclusions as the authors’ approach, it is not clear to me how important the assumption really is.

p5: More interestingly, we interpret this result independently of the somewhat-phenomenological
approach. We show that higher-rank MHD naturally arises as a consequence of the theory's
one-form symmetry when the conserved density corresponding to that one-form obeys certain global constraints.

The treatment in the paper is also somewhat phenomenological – please word more carefully. Does interpreting the result in terms of one-form symmetry change any physical predictions?

Conclusion: “The theories we have presented should provide the generic long time description of the quantum dynamics (at non-zero charge density) of fractonic phases of quantum matter most readily realized in experiment. Additionally, they should also provide the long time description for conventional spin liquid phases of quantum matter, where they have the advantage (relative to phenomenological approaches) of not relying on a weak coupling assumption.”

Please specify whether any new results are being claimed for conventional spin liquids, compared to existing predictions from phenomenological approaches. Also, the authors defer discussion of experiments to later work, but I am nevertheless curious which of the predictions in the paper they expect to be testable soonest (and in which material candidates etc.)

2. p8: “Generally speaking, sigma could be a matrix; however, we must construct sigma out of SO(3)-invariant objects: Since the only compatible such matrix is the identity, sigma reduces to a scalar”
Here it might be worth reminding the reader that SO(3) symmetry is being assumed throughout.

3. Sec. 2.3: It is not clear to me that SO(3) irreps are the most geometrically natural way to interpret this simplification – it looks like going from Eq. 2.20 to Eq. 2.22 is actually exploiting Hodge duality in the specific case that J_{ij} defines the components of a two-form. I am curious whether some of the new hydrodynamic equations elsewhere in the paper admit a similar geometrical formulation.

4. Eq. 2.29: I was a little confused by the arguments here based on time reversal, under which macroscopic reactive and dissipative terms generally behave differently. Perhaps oddness of the physical current under spatial inversion would be a more rigorous way to argue for allowed terms?
  • validity: -
  • significance: -
  • originality: -
  • clarity: -
  • formatting: -
  • grammar: -

Author:  Oliver Hart  on 2022-08-16  [id 2735]

(in reply to Report 2 on 2022-06-27)

The authors develop a theory of magnetohydrodynamics for the higher-rank gauge fields arising in fractonic systems. The core results of the paper are both timely and interesting, for example, the striking prediction of magnetic subdiffusion in fractonic MHD.

However, I did find some aspects of the presentation distracting, particularly the repeated claim that the approach in this work is fundamentally more rigorous and well-controlled than standard, "phenomenological" magnetohydrodynamics. I agree that the idea of Ref. 48, that flux conservation in MHD can be thought of as a slow mode arising from higher-form symmetry, is an appealing way to interpret the MHD equations. But it seems to me that the actual physical predictions in the paper could have been derived entirely within the phenomenological framework that the authors suggest is obsolete.

I therefore request that the authors either globally soften their language before publication, or better justify their claims.

We thank the referee for reviewing our manuscript and for their overall positive assessment of our work. The referee's points about our references to prior works (and comparisons to our work) are well taken. We have made changes throughout the manuscript to remedy the poor phrasing, which we describe in detail below and in the list of changes that will accompany our resubmission.

Importantly, we now characterize prior works not based on one-form symmetries as "semi-microscopic", rather than "phenomenological", and acknowledge that certain aspects of our manuscript (particularly the invocation of linear response) could be viewed as "phenomenological". Generally speaking, the equations of magnetohydrodynamics, as derived in classic literature, assume an approximate separability of the matter and electromagnetic stress tensors. This is, in practice, a weak coupling assumption—see also our response to the next point. An advantage of the symmetry-based approach, first emphasized in [PRD 95, 096003], is that one can use hydrodynamic principles to recover a coarse-grained theory of the long-time and long-wavelength dynamics of the fields themselves in the most physically relevant and interesting regimes, where fields are very strong and there may not be a clean separation between the matter and EM stress tensors. In general, the slow modes correspond to conserved quantities, and the advantage of our approach is that we derive our equations of motion using coarse-grained considerations of one-form symmetries, as opposed to a combination of microscopic equations.

While it is certainly possible in general for these two formulations to give equivalent results, in the semi-microscopic case, there is no intuition for why the long-wavelength physics one recovers ought to be universal. On the other hand, in the symmetry-based (effective theory) approach, this is quite transparent. Our derivation provides a valid description of the long-wavelength physics in arbitrary regimes, though equivalent results can be obtained using the semi-microscopic framework, as we show in a few cases (e.g., in sections 2.2.2, 3.2.2, and 4.3.2 in the updated manuscript). Alternatively, the symmetry-based approach tells us why the semi-microscopic approach behaves in the way that it does: because of the underlying higher-form symmetries. However, the existence of that symmetry is the only necessary prerequisite, and so while both approaches give rise to (sub)diffusion, the system need not be described by the exact semi-microscopic equations of motion to reach this same conclusion. Hence, the advantage of the "symmetry-based" approach—as opposed to the "semi-microscopic" approach—is primarily conceptual, but may also extend the regime of validity of the semi-microscopic approach (as opposed to predicting "new" results). We provide further details below in response to particular points made by the referee.

In particular, if there is anything in these results that could not have been derived using the standard method, it would help to make clear where the conclusions of the two approaches differ. If there is no material difference, it would be worth trying to address why the standard method works so well beyond weak coupling

For example, the discussion in the paper (and the claim in the earlier Ref. 48) has me wondering whether the standard derivation of MHD needs to invoke a weak-coupling assumption at all. Supposing the system has even one hydrodynamic mode (e.g. energy), one can coarse-grain the Maxwell equations to obtain a system of macroscopic Maxwell equations at the length and time scales associated with this hydrodynamic mode (using one of the many possible approaches in the literature). The linearization of these equations then yields linear response dynamics of the full system of matter+fields, and as usual, will predict conservation of magnetic flux density/decay of the other fields.

As indicated above, it is likely the case that similar or identical results could be derived using the semi-microscopic formalism. However, we believe our approach—rooted in symmetries and coarse graining—is better suited to the problem at hand, and gives reliable results in generic regimes. A useful analogy is to kinetic theory versus hydrodynamics: Although the two frameworks often agree (e.g., with respect to electron flow in wires), there are systems that are well described by hydrodynamics but not by kinetic theory. For many systems, including liquid water, kinetic theory is not accurate, as it fails to account for strong multi-particle correlations. Thus, there are conceptual advantages to using the hydrodynamic description from the outset in scenarios where it is valid (the regime of validity of hydrodynamics is well established in the literature), rather than using the more microscopic kinetic theory, even in situations where the two agree. In systems wherein the frameworks agree, they are formally related—and arguably equivalent; however, the hydrodynamic approach correctly captures the important physics of the longest-lived degrees of freedom in the system (corresponding to symmetries), and is capable of describing, e.g., liquid water, while kinetic theory is not. Thus, while we expect that equivalent results may be recovered using the semi-microscopic prescription of conventional MHD as applied to charged matter coupled to higher-rank gauge fields, we believe our approach to be more direct and conceptually more clear. It is also possible that recovering our particular results would require extra assumptions in the semi-microscopic approach, which are inherent to (and well justified) in the hydrodynamic approach we utilize.

This procedure works because the Maxwell equations are linear: the linearized, macroscopic Maxwell equations have the same mathematical form as the microscopic Maxwell equations. I wonder whether it is this aspect of the Maxwell theory (i.e. linearity) that means the phenomenological approach yields qualitatively correct predictions, rather than the weak-coupling assumption. (This might be one distinction from e.g. Yang-Mills.)

The procedure yields linearized Maxwell's equations because, within the hydrodynamic limit we consider, deviations from the linearized Maxwell's equations are subleading in the hydrodynamic expansion. However, this itself can be interpreted as a prediction of our hydrodynamic formalism since, in the context of emergent electromagnetism (or in the more familiar context of Maxwell's equations in matter, where, e.g., the polarization may depend nonlinearly on the electric field), the microscopic Maxwell's equations can be nonlinear. Part of the utility of our hydrodynamic formalism (as opposed to the semi-microscopic approach) is that the hydrodynamic coarse-graining procedure itself can be used to check that such nonlinear terms are irrelevant to the long-wavelength physics.

  1. On the question of standard MHD vs 1-form symmetries, please consider revising the following:

    "In contrast to phenomenological MHD, our approach based on higher-form symmetries provides a systematic treatment of hydrodynamics beyond weak coupling, and provides an effective hydrodynamic description, e.g., of quantum spin liquids, whose emergent gauge fields are strongly coupled."

I think this sentence overstates the level of rigor of the paper, as it is written. The approach of the authors is also phenomenological and e.g. uses the relaxation time approximation. It is also only "systematic" insofar as the one-form symmetry is exact, which is never true for emergent gauge fields as arise in condensed matter settings.

The sentence has been rephrased and toned down. The sentence now focuses, instead, on the fact that our work elucidates the origin of the (sub)diffusive behavior of the magnetic field tensor.

“Somewhat surprisingly, this was not carefully done even for ordinary electromagnetism until the past decade”

This seems to dismiss all the (vast) previous literature on this topic as "careless", which is unlikely to be the case. In particular, whether the magnetic slow mode is interpreted as local conservation of flux density or as a consequence of global 1-form symmetry is to some extent a matter of taste, and does not obviously depend on the coupling strength. Please consider revising.

We have remedied this poor phrasing by replacing "this was not carefully done even for ordinary electromagnetism" with "a first-principles derivation of magnetohydrodynamics using one-form symmetries was not done".

"This approach, based on higher-form symmetries, has significant conceptual advantages over more familiar phenomenological derivations of MHD. Specifically, whereas phenomenological derivations of MHD are limited in their validity to regimes wherein the gauge fields and matter are weakly coupled [48], the symmetry-based approach involving higher forms is valid for arbitrary coupling. In the context of emergent gauge theories in quantum spin liquids, there is no reason a priori to expect weak coupling between the matter and gauge fields [54]. An approach predicated on weak coupling is therefore suspect, and a more general approach—e.g., based on higher-form symmetries—is required...We emphasize that, while we focus on fractonic systems, our approach also applies to conventional spin liquids in the strongly coupled regime, where conventional phenomenological approaches may be uncontrolled."

I am not entirely convinced by this line of reasoning for the reasons discussed above. First, the higher-form symmetry is itself usually an approximation in physical realizations where alpha is believed to be large and the gauge field is emergent. Second, it appears that the part of the paper relevant to conventional spin liquids (i.e. Sec. II) recovers the conclusions of the conventional MHD approach, as taken in Ref. [62].

I therefore ask that the authors either soften their wording, or more carefully justify how leading-order, linear-response MHD requires the weak-coupling assumption—if this recovers the same conclusions as the authors' approach, it is not clear to me how important the assumption really is.

As we emphasized above, the main advantage of the new, "symmetry-based", perspective is primarily conceptual—it illustrates the underlying symmetries responsible for unusual diffusive or subdiffusive dynamics of the magnetic field tensor. Arguably, the underlying mechanism for this behavior would not have been understood without our perspective (as was the case for MHD as a theory with a one-form symmetry).

Nevertheless, we have toned down the wording used in the introduction, which we have modified to represent the advantages of our approach more accurately.

More interestingly, we interpret this result independently of the somewhat-phenomenological approach. We show that higher-rank MHD naturally arises as a consequence of the theory's one-form symmetry when the conserved density corresponding to that one-form obeys certain global constraints.

The treatment in the paper is also somewhat phenomenological – please word more carefully. Does interpreting the result in terms of one-form symmetry change any physical predictions?

In line with our reasoning given above, we have replaced "somewhat-phenomenological" with "semi-microscopic", which we believe better highlights the differences between the two approaches.

"The theories we have presented should provide the generic long time description of the quantum dynamics (at nonzero charge density) of fractonic phases of quantum matter most readily realized in experiment. Additionally, they should also provide the long time description for conventional spin liquid phases of quantum matter, where they have the advantage (relative to phenomenological approaches) of not relying on a weak coupling assumption."

Please specify whether any new results are being claimed for conventional spin liquids, compared to existing predictions from phenomenological approaches. Also, the authors defer discussion of experiments to later work, but I am nevertheless curious which of the predictions in the paper they expect to be testable soonest (and in which material candidates etc.)

In regards to material candidates, experimental realizations, and/or microscopic Hamiltonians, we refer the referee to our more detailed response to point 2. of referee 3. In short, the statement "most readily realized in experiment" that appeared in our conclusion referred to the implementation of gauged—rather than global—multipolar symmetries, which has been clarified in the updated paragraph.

The referee raises an interesting point regarding experimental signatures that may be derived from our results. We would like to stress that our manuscript is concerned with identifying and, importantly, understanding the universal behavior that derives from higher-rank Maxwell's equations (or a theory with a one-form symmetry obeying the constraints we enumerate). The consideration of experimental signatures thereof requires writing down a particular microscopic lattice model that gives rise to these equations of motion at long wave lengths. This would explain how the long-wavelength degrees of freedom are related to the microscopic ones, and hence, how the emergent higher-rank gauge fields couple to external (i.e., applied) fields that can be used to probe the system. Evidently, this step is predicated on the existence of candidate material realizations, and we hope that our work will provide motivation for identifying such candidate materials.

"Generally speaking, sigma could be a matrix; however, we must construct sigma out of SO(3)-invariant objects: Since the only compatible such matrix is the identity, sigma reduces to a scalar"

Here it might be worth reminding the reader that SO(3) symmetry is being assumed throughout.

We thank the referee for this suggestion. The sentence in question has been modified to remind the reader of this fact.

Sec. 2.3: It is not clear to me that SO(3) irreps are the most geometrically natural way to interpret this simplification – it looks like going from Eq. 2.20 to Eq. 2.22 is actually exploiting Hodge duality in the specific case that $J_{ij}$ defines the components of a two-form. I am curious whether some of the new hydrodynamic equations elsewhere in the paper admit a similar geometrical formulation.

We don't think it is likely that some clever applications of Hodge duality can explain the higher-rank theories we describe in a new geometric light. Indeed, after spending quite some time searching for one, we have presented the best possible perspective we could find in our manuscript (i.e., that the higher-rank diffusive models arise from adding certain constraints to a theory with a one-form symmetry). For example, by writing a theory with the charge density, $\rho_{ij}$, and the current, $J_{ij}$, both in the $\mathbf{5}$ of SO(3), we have already effectively used the Levi-Civita tensor to write both $\rho$ and $J$ as second rank tensors (and avoid writing $\partial_t \rho_{ij} + \partial_k J_{kij}$, subsequently assuming appropriate antisymmetry properties of $J_{kij}$), which is about as far as Hodge-duality can get you.

As far as we know, $p$-forms are the unique objects that can be naturally understood in a geometric light (e.g., one can define $\int \omega$ of a $p$-form $\omega$ over a $p$-dimensional surface without reference to coordinates). It is not obvious, for example, what, if any, is the natural object one should integrate a second-rank symmetric tensor over. In discussions with other experts in the community, this seems to be a major open problem relating, for example, to how/if one can naturally couple these higher-rank theories to gravity and curved space-time.

Eq. 2.29: I was a little confused by the arguments here based on time reversal, under which macroscopic reactive and dissipative terms generally behave differently. Perhaps oddness of the physical current under spatial inversion would be a more rigorous way to argue for allowed terms?

While oddness of the physical current under spatial inversion can be used to argue that the $J_i \propto \rho_i$ term is disallowed, we emphasize that the term is disallowed even when spatial inversion and/or time reversal symmetries are broken. This is seen most easily by noting that the equations of motion exhibit a thermodynamic instability, as we mention in the footnote. We have clarified in the main text that spatial inversion can alternatively be used to forbid $\alpha \neq 0$.

---

## Round 1 · Referee Report · Anonymous (Referee 3) · 2022-7-4

Strengths

The paper is pedagogical and easy to read

Weaknesses

The framework is introduced without a specific physical motivation

Report

The authors study diffusion of magnetic charges of the fraction type. Although the paper is an interesting generalisation of the previous developments there are several points that need to be addressed

  1. The authors refer to their framework as “fracton magnetohydrodynamics”. Traditionally magneto-fluid dynamics is the study of the magnetic properties and behavior of electrically conducting fluids i.e. systems that conserve momentum. On the other hand diffusion refers to the conservation of charge without momentum conservation. Equations considered in the paper do not conserve momentum and I do not see a reason to call the framework “magnetohydrodynamics” instead of diffusion. This is really confusing and dates back to the ref. [14] and subsequent works on the lattice models. At the time of ref. [14] no examples of hydrodynamic theories were known that conserved momentum. However, now this is not the case any more so it is even less justified to use the term hydrodynamics and not diffusion.

  2. The authors mention that they intend to describe spin liquids. However, no examples are given. It would be beneficial for the reader to have a more detailed discussion of a physical system, which can be described by the proposed formalism. By now there are several different examples of spin liquids and not all of them have the same emergent gauge fields.

  3. Several phenomenological relaxation times are assumed in the equations. However, given the emphasis on the symmetry arguments to constrain the form of equations, similar analysis should be introduced for these phenomenological coefficients.

  • validity: high
  • significance: ok
  • originality: low
  • clarity: high
  • formatting: excellent
  • grammar: excellent

Author:  Oliver Hart  on 2022-08-16  [id 2734]

(in reply to Report 3 on 2022-07-04)

We thank the referee for taking the time to review our manuscript. Their comments and criticisms are addressed in the point-by-point response below. A complete list of changes will accompany our resubmission.

  1. The authors refer to their framework as "fracton magnetohydrodynamics". Traditionally magneto-fluid dynamics is the study of the magnetic properties and behavior of electrically conducting fluids i.e. systems that conserve momentum. On the other hand diffusion refers to the conservation of charge without momentum conservation. Equations considered in the paper do not conserve momentum and I do not see a reason to call the framework "magnetohydrodynamics" instead of diffusion. This is really confusing and dates back to the ref. [14] and subsequent works on the lattice models. At the time of ref. [14] no examples of hydrodynamic theories were known that conserved momentum. However, now this is not the case any more so it is even less justified to use the term hydrodynamics and not diffusion.

We respectfully and fundamentally disagree with the referee's definition of "hydrodynamics": our modern understanding of hydrodynamics is that it is the coarse-grained theory that describes the relaxation of many-body systems to thermal equilibrium (or another stationary state). This is much broader than the Navier-Stokes equations, for example. This definition notwithstanding, the term "diffusion" is inappropriate in the context of our work since the theories that we consider may realize diffusion [Secs. 2, 3, 4] or subdiffusion [Sec. 5] of the magnetic field tensor in the presence of electrically charged matter.

While this more general application of the word "hydrodynamics" is now standard and broadly accepted in various communities, we acknowledge it is often unfamiliar to many readers. We have therefore opted to explain more clearly in the introduction the sense in which we use the word. See also the first point of our response to referee 1.

  1. The authors mention that they intend to describe spin liquids. However, no examples are given. It would be beneficial for the reader to have a more detailed discussion of a physical system, which can be described by the proposed formalism. By now there are several different examples of spin liquids and not all of them have the same emergent gauge fields.

Generally speaking, each section of our work applies to any microscopic quantum [or classical] system giving rise to emergent rank-$n$ electromagnetism that—in an appropriately taken continuum limit—is described by an SO(3)-symmetric, rank-$n$ variant of Maxwell's equations. We present the corresponding Maxwell's equations at the start of each section. The rank-one variant is well studied, and proposed realizations include numerous spin-liquid / spin-ice models that give rise to emergent compact electrodynamics, which we have cited more prominently in our manuscript.

For the rank-two case, there exist analogous lattice models that host rank-two gauge fields. These constructions are discussed, e.g., in [PRB 74 224433, arXiv:0602443, PRB 96 035119, PRB 98 165140]. Again, we have taken steps to make these citations more prominent at the beginning of Section 3.

The higher-rank case (i.e., for rank-three and above)—discussed in Section 4 of our manuscript—certainly has fewer proposed microscopic models in the literature than the rank-one and rank-two versions. To address the referee's comment, we have included a short section at the beginning of Section 4 outlining how to obtain higher-rank Maxwell's equations starting from a particular microscopic lattice model comprising O(2) quantum rotors.

Several phenomenological relaxation times are assumed in the equations. However, given the emphasis on the symmetry arguments to constrain the form of equations, similar analysis should be introduced for these phenomenological coefficients.

If $J_{ij}^{(\text{e})}$ and $E_{ij}$ transform as rank-two tensors under rotations, then, in general, the conductivity $\sigma_{ijkl}$ is a rank-four invariant tensor, which for SO(3) assumes the form $\sigma_{ijk\ell} = \alpha \delta_{ij}\delta_{k\ell} + \beta \delta_{ik}\delta_{j\ell} + \gamma \delta_{i\ell}\delta_{jk}$. If $E_{ij}$ is symmetric and traceless, as is the case in section 3, then the relation between current and electric field tensor reduces to $J_{ij}^{(\text{e})} = \sigma_e E_{ij}$. That is, the conductivity tensor reduces to a scalar quantity $\sigma_e$ that cannot be further constrained by symmetry. The same is true for the rank-$n$ symmetric tensors considered in Sec. 4.

The same is not true for the traceless vector charge theory presented in Sec. 5. In the most general setting permitted by symmetry, since the electric field tensor transforms in a reducible representation, the different irreps can be associated with different decay rates. This leads to a generalization of the quasi-normal mode structure in which the modes $B_{xx} = \pm B_{yy}$ have differing subdiffusion constants.

---

## Round 2 · Referee Report · Anonymous (Referee 1) · 2022-9-1

Report

The authors answered my questions satisfactorily. I agree with referee three that the paper lacks concrete physical systems. However, the article provides a theoretical prescription for higher-rank hydrodynamics. I recommend this paper be published on SciPost Physics.

---

## Round 2 · Referee Report · Anonymous (Referee 2) · 2022-9-7

Report

I thank the authors for their detailed and thoughtful response. The revised manuscript seems to provide a clearer account of what was achieved, and I am happy to recommend it for publication.

---

## Round 2 · Referee Report · Anonymous (Referee 3) · 2022-9-8

Report

I thank the authors for addressing my points. However, I still think the revised version does not convincingly justify the use of the name “fracton magnetohydrodynamics” in the title. Even in the view of modern developments I don’t think diffusion is referred to as hydrodynamics. For example in the recent book Relativistic Fluid Dynamics In and Out of Equilibrium by Paul Romatschke and Ulrike Romatschke diffusion is referred to as an analogy to fluid dynamics. Also modern developments in magnetohydrodynamics based on two-form symmetries focus on systems with momentum conservation. Fractons admit a hydrodynamic description with and without momentum conservation. This leads to the conclusion that calling diffusion or sub-diffusion (magneto)hydrodynamics leads to confusion. This is the case even if the authors can write in the text what they really mean. As a result I think that the title suggests more than the authors actually do and, in my opinion it has to be changed, to avoid misleading the reader. If the authors think that using the term magnetic diffusion is inappropriate they can suggest another name but certainly not “fracton magnetohydrodynamics”.

Requested changes

  1. Change the tile and possibly avoid abusing the term magnetohydrodynamics.

---

## Round 2 · Author Response

We thank the referees for their careful reading and detailed feedback on our manuscript, and the editor for arranging this review of our manuscript. We have provided a point-by-point response to each of the referees on the submission page. A detailed list of the corresponding changes made to the manuscript in response to the referees' feedback is given below. We hope that these revisions have satisfied the referees' concerns and that the current manuscript will be judged suitable for publication in SciPost Physics.

---

## Round 2 · List of Changes

In response to Report 1:

1) the first paragraph of the introduction has been modified to clarify our use of the term "hydrodynamics"

> "the universal properties of these theories can be characterized within the framework of hydrodynamics, which is the coarse-grained effective theory of the long-time and long-wavelength dynamics of systems as they relax to equilibrium"

2) Additionally, we have added a footnote after the sentence ending "...these long-lived modes are associated with conserved densities (or Goldstone bosons), and their dynamics is dubbed "hydrodynamics""

> "Note that our use of the term "hydrodynamics"—the coarse-grained description of systems as they relax to equilibrium—does not require that momentum be conserved, and need not correspond to the Navier-Stokes equations, for example."

3) Above Eq. (3.6), we have explained that the current tensor and the electric field tensor being proportional to one another is, in fact, the most general form of Ohm's law permitted by symmetry, if the electric field tensor transforms in the 5 of SO(3).

4) Similarly, below Eq. (4.7), we have made analogous arguments for the rank-n case, where the electric field tensor is a symmetric, traceless, rank-n quantity.

5) For the traceful scalar charge theory in Sec. 5, Eqs. (5.5) through (5.7) have been modified to include the two time scales that are in general permitted by SO(3) rotational invariance, since the trace part and the traceless symmetric part can decay on different time scales.

6) We have added the following sentence below Eq. (3.10) to remind the reader that the vanishing of magnetic charge is only ever an approximation in emergent theories:

> "We remind the reader that the vanishing of magnetic charge density can at best only be expected to hold approximately in emergent theories; see Sec. 3.2.3 for a discussion of the length and time scales over which (3.10) provides an accurate description of the dynamics."

7) Below Eq. (3.13) we have added the following sentences to clarify that the structure of time-independent gauge transformations and the conserved moments of the magnetic field tensor coincide only in the case of self-dual theories:

> "It is worth noting that (3.13) coincides with the structure of time-independent gauge transformations acting on the vector potential $A_{ij}$, canonically conjugate to $E_{ij}$. This apparent equivalence derives from the self-dual nature of the traceless scalar charge theory—i.e., the derivative and tensor structure of the electric and magnetic Gauss’s laws is identical."

8) The first sentence of Sec. 2.2.5 has been modified in order to define the "magnetic conductivity".

9) The tilde has been removed in (2.21) and in the subsequent paragraph.

10) The sentence in the introduction that states the result for the diffusion constant for rank-n theories has been modified to clarify that the result applies only to rank-n traceless symmetric theories.

In response to Report 2:

11) The abstract has been modified to better represent the utility of a symmetry-based approach:

> "In contrast to semi-microscopic derivations of MHD, our approach elucidates the origin of the hydrodynamic modes by identifying the corresponding higher-form symmetries. Being rooted in symmetries, the hydrodynamic modes may persist even when the semi-microscopic equations no longer provide an accurate description of the system."

12) Similarly, in the introduction, we have emphasized that the main advantage of the "symmetry-based" approach is primarily conceptual in nature:

> "Somewhat surprisingly, a first-principles derivation of magnetohydrodynamics using one-form symmetries was not done until the past decade..."

and

> "This approach, based on higher-form symmetries, has significant conceptual advantages over more familiar semi-microscopic derivations of MHD. Specifically, the symmetry-based approach highlights the underlying symmetries responsible for the observed long-wavelength modes, while also being less limited in its regime of validity than the semi-microscopic approach. For example, in conventional (rank-one) MHD, the semi-microscopic derivation invokes approximate separability of the electromagnetic and matter stress tensors. In the symmetry-based approach [48], one invokes hydrodynamic principles to recover a coarse-grained theory of the long-time and long-wavelength dynamics of the fields in the most interesting and physically relevant regimes, where there may not be a clean separation between the two tensors. This approach also gives predictions for particular limits of conventional U(1) spin-liquids in which the relevant symmetries are weakly broken. In the case of emergent electromagnetism in fractonic spin liquids, the emergent gauge fields are higher rank, leading to additional subtleties and new universality classes. The hydrodynamic description of these higher-rank theories is the subject of this work."

13) In the conclusion,

> "The theories we present describe the generic long-time description of the quantum dynamics of fractonic phases (at nonzero charge density) exhibiting gauged—as opposed to global—multipolar symmetries (as relevant to spin liquids and fracton phases). We develop a symmetry-based approach describing arbitrary higher-rank theories of this type, and showcase the subdiffusion of magnetic fields as an example of the exotic universal dynamics that may arise in this context."

14) At the top of page 9, we have reminded the reader that SO(3) rotational invariance is assumed throughout the manuscript.

15) Below (2.29), we have explained that spatial inversion can alternatively be used to forbid the term proportional to the vector density, i.e., to require $\alpha = 0$.

In response to Report 3:

[See points 1) and 2) in response to Report 1, which address the manner in which we use the term "hydrodynamics"]

16) To highlight the physical systems that our results apply to, at the start of each section we have made more prominent the citations that discuss concrete examples of lattice models giving rise to emergent electromagnetism of the corresponding rank.

17) Since there are few examples in the literature, we have added a new subsection [Section 4.2] in which we illustrate an emergent rank-three variant of QED in a lattice O(2) quantum rotor system.

18) We have also included an additional figure (Fig. 2) to accompany the introduction of the rank-three lattice model.

[See points 3), 4), 5) in response to Report 1, which employ a symmetry-based analysis to constrain the form of Ohm's law]

---

## Editorial Decision

resubmitted